# MSH2 stimulates interfering and inhibits non-interfering crossovers in response to genetic polymorphism

Julia Dluzewska [1], Wojciech Dziegielewski [1], Maja Szymanska-Lejman [1], Monika Gazecka [1,4], Ian R. Henderson [2], James D. Higgins [3] & Piotr A. Ziolkowski [1]✉

Meiotic crossovers can be formed through the interfering pathway, in which one crossover prevents another from forming nearby, or by an independent non-interfering pathway. In Arabidopsis, local sequence polymorphism between homologs can stimulate interfering crossovers in a MSH2-dependent manner. To understand how MSH2 regulates crossovers formed by the two pathways, we combined Arabidopsis mutants that elevate non-interfering crossovers with *msh2* mutants. We demonstrate that MSH2 blocks non-interfering crossovers at polymorphic loci, which is the opposite effect to interfering crossovers. We also observe MSH2-independent crossover inhibition at highly polymorphic sites. We measure recombination along the chromosome arms in lines differing in patterns of heterozygosity and observe a MSH2-dependent crossover increase at the boundaries between heterozygous and homozygous regions. Here, we show that MSH2 is a master regulator of meiotic DSB repair in Arabidopsis, with antagonistic effects on interfering and non-interfering crossovers, which shapes the crossover landscape in relation to interhomolog polymorphism.

Sexual reproduction involves the fusion of gametes formed by a specific cell division called meiosis[1,2]. During meiosis, homologous chromosomes pair and exchange genetic information through crossover[2,3]. Crossovers are initiated by programmed DNA double-strand breaks (DSBs), most of which are repaired as non-crossovers[1–3]. Crossovers create new allelic combinations, which are crucial for adaptation, diversity and evolution[4–6]. Moreover, they are necessary for proper chromosome segregation in meiosis, meaning non-recombining mutants are sterile[2,3,7,8]. In most eukaryotes, two crossover classes exist, that are produced by different pathways[9,10]. Class I crossovers arise via the ZMM pathway, which is named after the recombination proteins ZIP1-4, MSH4/5 and MER3. Class I crossovers are interfering, so that one crossover inhibits the formation of another in a distance-dependent manner[11,12]. In Arabidopsis, crossover interference depends

on the formation of a synaptonemal complex, which is a proteinaceous structure that assembles between homologous chromosomes[13–15]. The number of Class I crossovers is limited by the level of HEI10 expression, which is a ZMM protein with a function of E3 ubiquitin/SUMO ligase[16–18]. In contrast, non-interfering Class II crossovers are dependent on structure-specific nucleases including MUS81 and are strongly inhibited by DNA helicases, mainly FANCM and RECQ4A/RECQ4B in plants[19–27]. Therefore, the number of crossovers in both pathways is kept relatively low, e.g., ~8 Class I and ~1 Class II crossovers per meiosis in *Arabidopsis thaliana*[28–30].

The distribution of crossovers along the chromosomes is not uniform and is largely determined by the chromatin state[31–41]. DNA polymorphisms between homologous chromosomes can also affect crossover placement[42–47]. For example, the juxtaposition of

[1]Laboratory of Genome Biology, Institute of Molecular Biology and Biotechnology, Adam Mickiewicz University, Poznań, Poland. [2]Department of Plant Sciences, University of Cambridge, Cambridge, UK. [3]Department of Genetics and Genome Biology, University of Leicester, Leicester, UK. [4]Present address: Department of Molecular Virology, Institute of Bioorganic Chemistry, Polish Academy of Sciences, Poznań, Poland. ✉e-mail: pzio@amu.edu.pl

heterozygous and homozygous regions stimulates crossover in the heterozygous region[48]. This effect is specific to the ZMM pathway and also depends on MSH2[49,50]. MSH2 is a key subunit of complexes that detect base mismatches occurring in somatic cells, as a consequence of replication errors[51–53]. In meiosis, MSH2 complexes detect mismatches in heteroduplexes, which are formed during strand invasion initiated in heterozygous regions[54–56]. Interestingly, genome-wide comparison of crossover distributions between hybrids, obtained by crossing two genetically diverged *A. thaliana* accessions, and quasi-inbreds, obtained by crossing homozygous inbred lines in which a few hundred genetic markers were introduced through mutagenesis, showed no significant differences[40].

In this work, we further explore the relationship between polymorphism and meiotic recombination to investigate the apparent contradiction between crossover measurements made for individual chromosome regions versus genome-wide data along entire chromosomes[40,48,49]. To this end, we examine crossover formation in knock-out *msh2* mutants depending on the activity of interfering versus non-interfering crossover pathways. We applied both approaches based on genome-wide crossover maps, as well as recombination analysis using fluorescent reporter lines (FTLs) that measure individual chromosome regions. We show that inactivation of MSH2 in *fancm* or *recq4* backgrounds leads to a significant increase in Class II crossovers in *A. thaliana* hybrids. Furthermore, we demonstrate that polymorphism inhibits Class II crossovers via both MSH2-dependent and independent mechanisms. By analysing crossovers using FTLs that cover the entire left arm of chromosome 3, we showed that crossover distributions are very similar between inbred and hybrid backgrounds. However, a change in the local pattern of heterozygosity along the chromosome induces a dramatic Class I crossover redistribution, with crossover increasing across the heterozygous/homozygous boundary. It seems that while in plants Class I crossovers function to secure high genetic variation through the recombination between genetically non-identical homologous chromosomes, the main role of Class II crossovers is to safely eliminate any unrepaired DSBs that may be dangerous for genome integrity[6]. Therefore, we propose that differences in the effect of MSH2 on the Class I and II crossovers in response to interhomolog polymorphism are a consequence of distinct biological functions of the two pathways.

## Results

### MSH2 limits fertility of *fancm zip4* hybrids

Inactivation of the *ZIP4* gene in Arabidopsis leads to a complete blockage of the ZMM pathway responsible for Class I crossovers and, consequently, a strong reduction of plant fertility due to crossover scarcity[29]. Earlier studies have shown that a knock-out mutation in the *FANCM* gene restores the fertility of *zip4* mutants by elevating non-interfering Class II crossovers in homozygous backgrounds of the *A. thaliana* accession, Columbia (hereafter Col)[20]. However, restoration of *zip4* fertility by *fancm* is largely inhibited in hybrids between different accessions, for example, Col and Landsberg *erecta* (hereafter L*er*), which differ by an average of 6.7 SNPs per kb[24,48,57]. To confirm differences in fertility between *fancm zip4* Col or L*er* inbreds, and *fancm zip4* Col/L*er* hybrids, we first generated a knock-out *zip4* mutation in the *fancm* L*er* background using CRISPR-Cas9 (Supplementary Fig. 1). The L*er fancm zip4* plants were fully fertile, similarly to their Col counterparts (Supplementary Figs. 1 and 2). By crossing Col *fancm zip4* with L*er fancm zip4*, we obtained hybrid plants which showed reduced fertility (Fig. 1a–d). Therefore, we hypothesised that the low fertility of the *fancm zip4* Col/L*er* hybrid is due to limited crossover recombination, which in turn is a consequence of the DNA polymorphism between homologues.

In eukaryotes, heterodimers of bacterial MutS homologues (MSHs) that bind mismatched bases are responsible for detecting interhomolog polymorphism[54,58,59]. All MSH dimers involved in

mismatch detection are assumed to contain the MSH2 protein[49,50,54,58–60]. Therefore, we investigated whether inactivation of *MSH2* would increase the fertility of *fancm zip4* Col/L*er* hybrids. We obtained triple mutants by backcrossing *msh2* to *fancm zip4* double mutants in both Col and L*er* backgrounds. We observed that the triple *msh2 fancm zip4* hybrid produced more seeds than the *fancm zip4* hybrid (Fig. 1a–d). While *msh2 fancm zip4* plants showed a lower number of seeds per silique (mean 20.1) than wild type (mean 56.7), it was significantly higher than *fancm zip4* (mean 7.5, Welch's *t* test $P = 4.2 \times 10^{-5}$) (Fig. 1c). Based on these findings, we concluded that the fertility of *fancm zip4* hybrids can be partially rescued by inactivation of the *MSH2* gene.

### Class II crossovers are repressed by MSH2 in the *fancm* hybrids

The absence of at least one crossover per chromosome pair prevents the dissociation of bivalents into univalents after the disassembly of the synaptonemal complex. Consequently, in metaphase I, apart from bivalents in the shape of rings (indicating at least two chiasmata) and rods (indicating one chiasma), univalents are observed[61]. To examine whether the increased fertility of the *msh2 fancm zip4* hybrid was associated with an increase in crossover numbers, we analysed chromosome cytological configurations in meiotic metaphase I (Fig. 1e, f). In *msh2 fancm zip4*, 1.40 pairs of univalents per cell were observed compared to 3.47 in *fancm zip4* (Mann–Whitney test $P = 4.42 \times 10^{-10}$). The number of ring bivalents per cell increased from 0.12 in *fancm zip4* to 0.87 in *msh2 fancm zip4* (Mann–Whitney test $P = 4.12 \times 10^{-9}$). These changes coincided with a 2.8-fold increase in chiasmata per cell, from 1.40 in *fancm zip4*, to 3.88 in *msh2 fancm zip4* (Mann–Whitney test $P = 2.87 \times 10^{-14}$; Supplementary Fig. 3). This result shows that *msh2* increases chiasma formation in *fancm zip4* hybrids.

Segregation of linked T-DNAs that express different colours of fluorescent protein (fluorescent-tagged lines, FTLs) can be used to measure crossover frequency within defined chromosomal intervals (Fig. 1h)[62,63]. The Col-*420* line (Col background) is an FTL, in which seed-expressed T-DNAs encoding eGFP and dsRED define the *420* interval spanning 5.1 Mb close to the telomere of chromosome 3[48,62]. Therefore, we backcrossed combinations of *msh2*, *fancm* and *zip4* mutants to the Col-*420* reporter line, and then obtained F$_1$ plants via crosses to the corresponding mutants in L*er*. Since fertility in *zip4* is drastically reduced by low crossovers, the seed produced in these mutants usually results from gametes that experienced more crossovers than the mutant average[29]. Thus, crossover measurements will be overestimated, especially in the *fancm zip4* mutant. Comparing *420* crossover frequency between *fancm zip4* and *msh2 fancm zip4* showed a statistically significant increase (Welch's *t* test $P = 8.6 \times 10^{-9}$; Fig. 1g). Based on these observations, we concluded that MSH2 inhibits Class II crossovers in *fancm zip4* hybrids.

Although chiasma numbers in *msh2 fancm zip4* are much lower than wild type, crossover measurements in the *420* interval show the opposite effect. One of the reasons is likely the above-mentioned overestimation of the crossover frequency measured in seeds in lines carrying the *zip4* mutation. However, it is also possible that the discrepancy between chiasma number and *420* crossover measurements is due to uneven crossover distribution along the chromosomes, with a higher frequency of crossovers in the sub-telomeric regions (like *420*), and lower frequency in interstitial and pericentromeric regions. Therefore, we next investigated crossover distributions on a genome-wide scale. We generated F$_2$ populations from Col × L*er* crosses in the genetic background of *fancm zip4*, *msh2 fancm*, and *msh2 fancm zip4* mutants. We sequenced genomic DNA from between 175 and 211 F$_2$ individuals from each population and identified crossovers in each individual (Fig. 2a, b and Supplementary Table 3). Crossovers were identified as genotype switches along the chromosomes and were assigned to the midpoint between pairs of SNPs[64]. On this basis, we obtained data on the number and location of crossovers in the tested

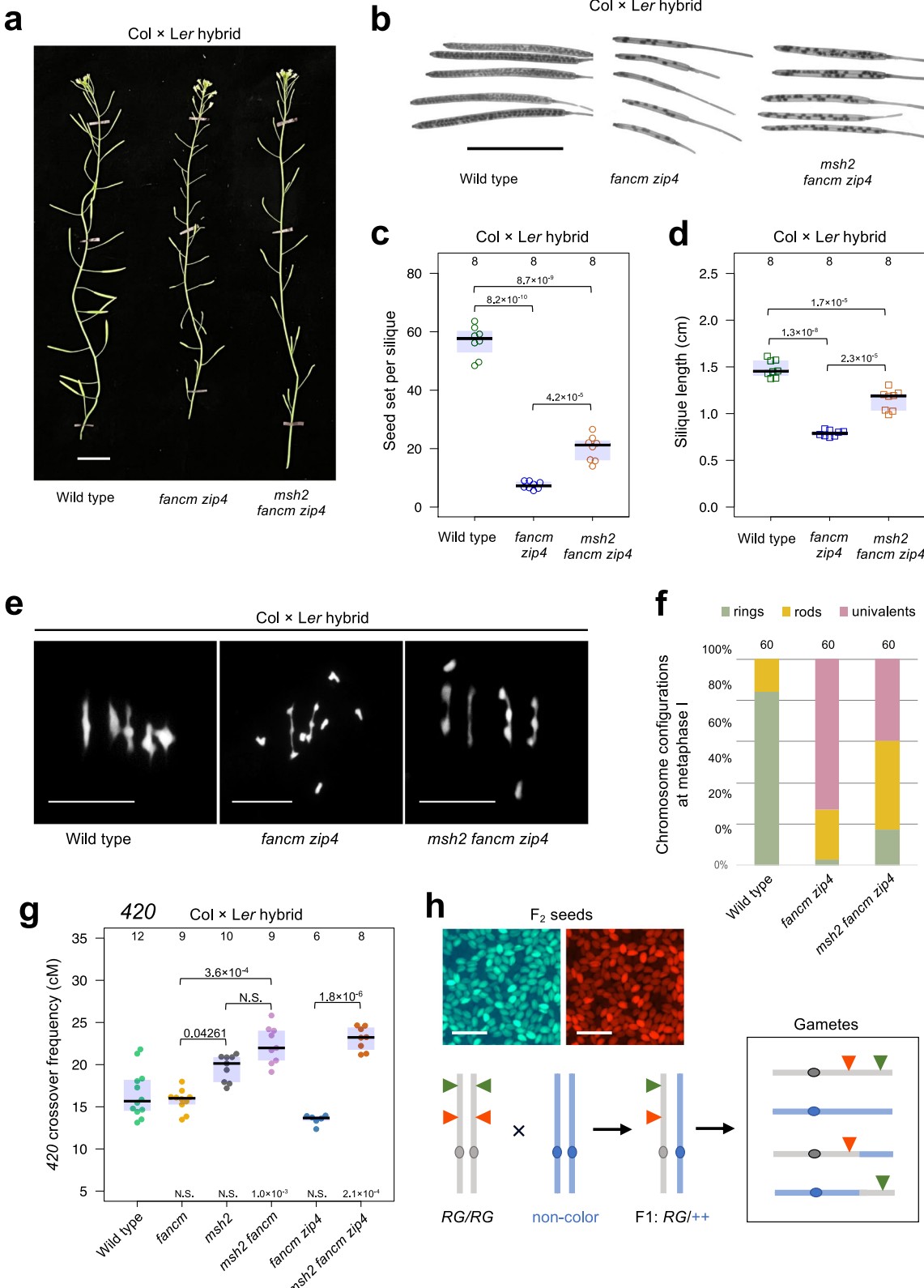

mutants, which we compared with the analogous data for wild-type and previously published *msh2* Col × L*er* crosses[49] (Fig. 2). Both *fancm zip4* and *msh2 fancm zip4* showed lower crossover numbers than wild type (Welch's *t* test $P = 4.2 \times 10^{-14}$ and $P = 4.6 \times 10^{-4}$, respectively; Fig. 2b). Moreover, *msh2 fancm zip4* showed significantly higher crossovers than *fancm zip4*, confirming our previous observations

based on chiasma counts ($P = 2.2 \times 10^{-16}$; Figs. 1f and 2b). We also observed elevated crossovers in *msh2 fancm* when compared to wild type (11.0 versus 8.0 crossovers per $F_2$, $P = 1.1 \times 10^{-11}$). Previous reports indicated that crossover frequency in *fancm* Col/Col inbreds is two to three times higher than in wild type, due to increased Class I crossover numbers, while Col/L*er* hybrids are not significantly higher

**Fig. 1 | Inactivation of *MSH2* partially restores fertility in *fancm zip4* Col × L*er* hybrids. a** Representative images of wild-type, *fancm zip4* and *msh2 fancm zip4* Col × L*er* hybrids. Scale bar, 2 cm. **b** Representative cleared siliques of wild-type, *fancm zip4* and *msh2 fancm zip4* Col × L*er* hybrids. Scale bar, 1 cm. **c, d** Fertility assays in Col × L*er* *fancm zip4* and *msh2 fancm zip4* as assessed via seed set (**c**) and silique length (**d**). The centre line of a boxplot indicates the mean; the upper and lower bounds indicate the 75th and 25th percentiles, respectively. Each dot represents a measurement from five siliques of one plant. The numbers of individuals are indicated above the boxplots. The two-sided *P* values were estimated by Welch's *t* test. **e** DAPI-stained metaphase I chromosome spreads from Col × L*er* male meiocytes in wild type, *fancm zip4* and *msh2 fancm zip4*. Scale bars, 5 μm. **f** Proportions of rings, rods and univalents per Col × L*er* male meiocyte in wild type, *fancm zip4* and *msh2 fancm zip4*. Three individual plants per genotype were scored. The total number of cells sampled for each genotype is indicated above the bars.

**g** *420* crossover frequency (cM) in Col × L*er* hybrids of *fancm*, *msh2*, *msh2 fancm*, *fancm zip4* and *msh2 fancm zip4*. The two-sided *P* values were estimated by Welch's *t* test. The centre line of a boxplot indicates the mean; the upper and lower bounds indicate the 75th and 25th percentiles, respectively. Each dot represents a measurement from one individual. The numbers of individuals are indicated above the boxplots. **h** Genetic diagram illustrating the seed scoring approach with a single chromosome pair shown for simplicity. Fluorescent reporters in FTL are indicated as green and red triangles. FTL in the Col background (grey) is crossed to L*er* (blue) to generate F₁ hybrids. Following meiosis the proportion of parental:crossover gametes from F₁ is analysed to measure genetic distance (cM) between the fluorescent protein-encoding transgenes. Top panel shows the representative photographs of seed fluorescence segregation in *420* interval. Scale bars, 3 mm. Source data are provided as a Source Data file.

than in wild type[20,24]. Therefore, it can be assumed that the inactivation of *MSH2* in *fancm* also leads to an increase in Class II crossovers (Fig. 2b), although effects of FANCM inactivation on Class I crossover distributions cannot be excluded[25,65]. Together, our results indicate that *MSH2* inactivation increases Class II crossover frequency in the range blocked by the FANCM helicase in Col/L*er* hybrids.

We then compared crossover distributions along the chromosomes (Fig. 2c–e). The crossover profiles for *fancm zip4* and *msh2 fancm zip4* showed a reduced frequency of recombination in pericentromeric regions compared to wild type (Fig. 2c–e and Supplementary Fig. 4). Moreover, the *fancm zip4* and *msh2 fancm zip4* crossover distributions were strongly correlated (Spearman *Rho* = 0.701, *P* < 2.2 × 10⁻¹⁶; Fig. 2c). In turn, the crossover distribution along chromosomes for the *msh2 fancm* double mutant was intermediate between the pattern observed in the *msh2 fancm zip4* triple mutant (Spearman Rho = 0.752, *P* < 2.2 × 10⁻¹⁶; Fig. 2c) and wild type (Rho = 0.434, *P* < 2.2 × 10⁻¹⁶; Fig. 2c). In both *msh2 fancm zip4* and *msh2 fancm*, we observed an increased proportion of crossovers close to the chromosome ends compared to wild type (Fig. 2d, e and Supplementary Fig. 4). These results show that Class II crossovers are repressed by FANCM primarily in the sub-telomeric regions.

## Class II crossovers are repressed by MSH2 in *recq4a recq4b* hybrids

In contrast to the *fancm* mutation, in which an increase in recombination frequency is observed only in inbreds, the *recq4a recq4b* mutants (hereafter *recq4*) show dramatically elevated non-interfering Class II crossover level in both inbreds and hybrids[20,23,24]. However, the increase observed in *recq4* hybrids is always lower than that observed in a fully homozygous background[24,66]. Therefore, we investigated whether MSH2 limits crossover in *recq4* hybrid plants. For this purpose, we obtained *msh2 recq4* mutants in the Col/Col, L*er*/L*er* and Col/L*er* backgrounds. Then, we sequenced 279 F₂ individuals derived from *msh2 recq4* Col/L*er* hybrids and identified crossover sites. The *msh2 recq4* plants showed an average of 30 crossovers per individual, which is significantly greater than the 23 crossovers observed in the *recq4* double mutant alone (Welch's *t* test *P* < 2.2 × 10⁻¹⁶) (Fig. 2b). This suggests that MSH2 represses Class II crossover formation in response to interhomolog polymorphism in the pathway inhibited by RECQ4 helicase.

However, the crossover distribution along chromosomes was strongly correlated between *msh2 recq4* and *recq4* genotypes (Spearman *Rho* = 0.958, *P* < 2.2 × 10⁻¹⁶, Fig. 2b–d). While an increase in crossover frequency was observed primarily along the chromosome arms and in proximity to the chromosome ends, the difference between the two genotypes was small in the centromere-proximal regions (Fig. 2d, e). Therefore, we conclude that MSH2-dependent polymorphism detection has a limited effect on Class II crossover formation in the pericentromeres, at least in *recq4* hybrid background.

## MSH2 has the opposite effect on the Class I and Class II crossovers in response to interhomolog polymorphism

To investigate the extent that SNPs affect crossover activity in individual mutants, we divided the genome into 100 kb non-overlapping windows for which we determined SNP density and crossover frequency using genome-wide data generated in this work as well as published previously[41,49,66]. This resulted in 1191 windows, which we sorted according to the SNP density and divided them into 99 groups of SNP density windows (Supplementary Fig. 5). We plotted the relationship between SNP density and crossover frequency for wild type (Fig. 3a). This revealed a parabolic relationship, which we described previously[49]. We then plotted the same relationship for the mutants, but after normalising to wild type. Specifically, by subtracting crossover frequency for wild type from the given mutant, for each SNP density group.

In '*msh2*−wild type', we observed a modest decrease in crossover frequency in almost all SNP density groups (Fig. 3b). Stronger reductions in crossovers were observed in '*fancm zip4*−wild type' and '*msh2 fancm zip4*−wild type', but only for SNP density groups with a high density of SNPs: below 2.5 ('*fancm zip4*−wild type') and 5 SNPs/kb ('*msh2 fancm zip4*−wild type') there is a noticeable increase in crossover frequency (Fig. 3c, d). Despite the inactive ZMM pathway in both mutants caused by *zip4*, we did not observe a decrease in crossover frequency in SNP density groups with a high density of SNPs (>18 SNPs/kb), as these regions are close to being recombinantly inactive in wild type (Fig. 3a). The analysis of '*msh2 fancm*−wild type' shows an increase in crossover frequency that is negatively proportional to SNP density in groups below ~8 SNPs/kb, and a decrease in crossovers above this limit of SNP density (Fig. 3e). Altogether, these data show that Class II crossovers generated in the *fancm* mutant are strongly inhibited even at relatively low SNP densities, and *msh2* inactivation increases their tolerance to SNPs to some extent.

In both '*recq4*−wild type' and '*msh2 recq4*−wild type', the relationship between SNP density and crossover frequency show a more complex trend with an apparent peak at ~3–4 SNPs/kb (Fig. 3f, g). This may indicate that Class II crossovers formed in *recq4* are less sensitive to interhomolog polymorphism and most often occur in regions containing 3–4 SNPs/kb. Below 4 SNPs/kb in both mutants we observe a decrease in the frequency of crossovers proportional to SNP density, which reached the wild-type level at approximately >15 SNP/kb (Fig. 3f, g). Although the plots of the relationship between SNP density and crossover frequency for '*recq4*−wild type' and '*msh2 recq4*−wild type' look similar, the latter mutant shows overall greater increases in crossover frequency compared to wild type. This suggests that MSH2 inactivation increases crossover activity in *recq4* (Fig. 3f, g).

To explore the difference in Class II crossovers made by the *msh2* mutation, we subtracted *recq4* crossover frequency from *msh2 recq4* frequency for each SNP density group ('*msh2 recq4* − *recq4*'; Fig. 3i). We observed that the increase in crossover frequency in the absence of

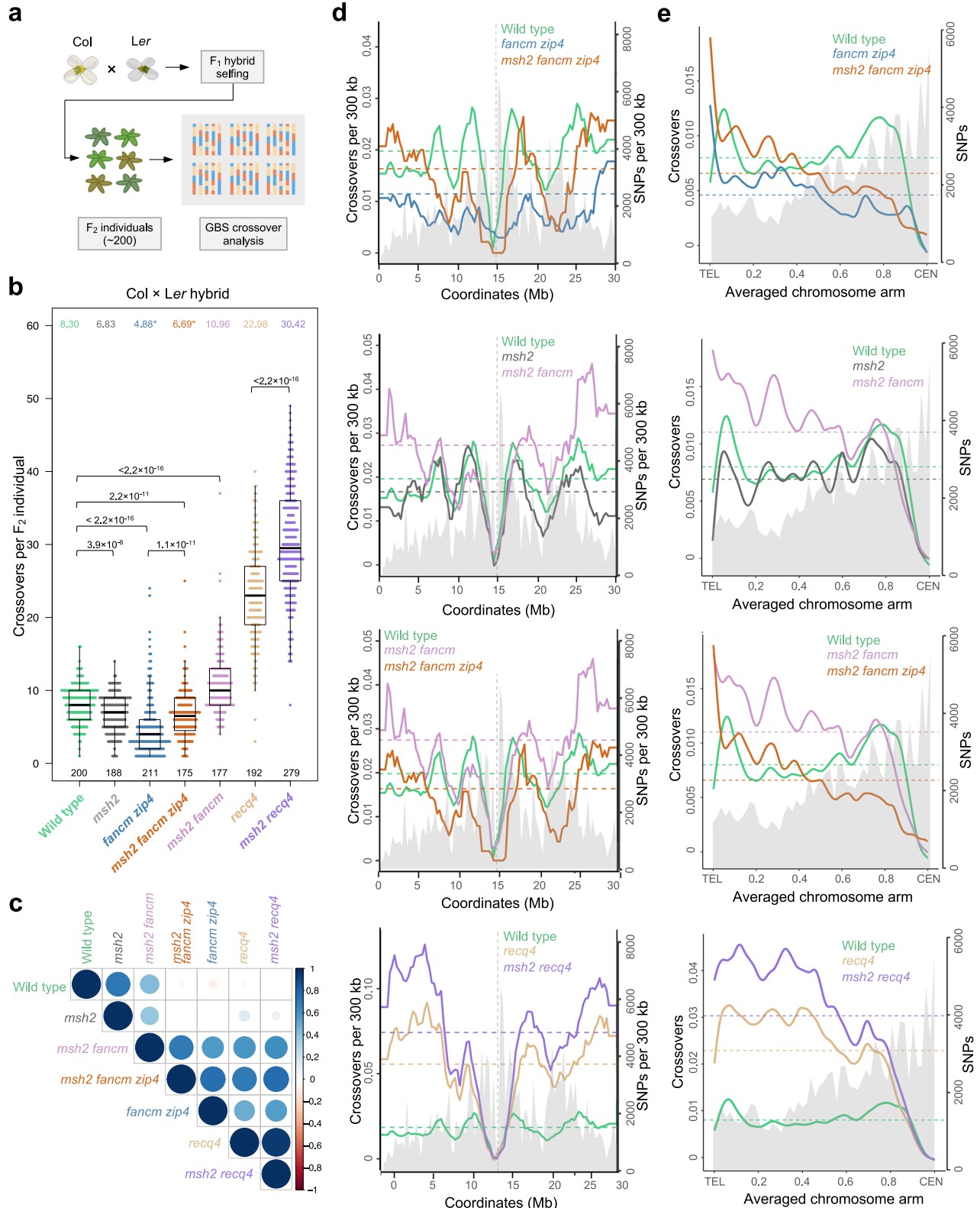

MSH2 occurs in regions with a large spectrum of SNP densities (from >0 to <20 SNPs/kb) but is greatest at ~4 SNPs/kb. The effect of MSH2 inactivation is especially strong in regions with relatively low SNP density (~2–6 SNPs/kb), where it leads to an increase in crossover frequency by up to 2 cM/100 kb (Fig. 3i). The same analysis for 'msh2 fancm zip4 – fancm zip4' showed a similar, though weaker relationship between SNP density and crossover frequency change (*Rho* = −0.314, *P* = 0.0016; Fig. 3h). Together, our data for *recq4* and *fancm zip4*

suggest that MSH2 effectively inhibits Class II crossovers in the range of 2–6 SNPs/kb, while outside this range its effect is much smaller.

A comparison of the analyses for 'msh2–wild type' (Fig. 3b), versus 'msh2 fancm zip4–fancm zip4', and 'msh2 recq4–recq4' (Fig. 3h, i) suggests that MSH2 affects crossovers differently in Class I and Class II pathways. In the Class I pathway, MSH2 stimulates recombination, while in pathways leading to the formation of Class II crossovers, MSH2 inhibits recombination. The MSH2 inhibition effect,

**Fig. 2 | The effect on *MSH2* inactivation on crossover frequency and distribution in different mutant backgrounds. a** Diagram illustrating crossover mapping in F$_1$ plants based on F$_2$ individuals. **b** The number of crossovers per F$_2$ individual in the indicated populations. For wild type, 200 randomly selected individuals were plotted. Significance was assessed by Kruskal–Wallis H test followed by Mann–Whitney *U* test with Bonferroni correction. The centre line of a boxplot indicates the mean; the upper and lower bounds indicate the 75th and 25th percentiles, respectively; the whiskers indicate the minimum and maximum. The mean is indicated also on the top. Asterisks by means denote genotypes where crossover numbers are likely to be inflated due to reduced fertility (only gametes with sufficient crossover numbers can form the sequenced F$_2$ generation). The numbers of individuals are indicated below the boxplots. **c** The correlation coefficient matrices among genome-wide crossover distributions as calculated in 0.3 Mb adjacent windows. **d** Crossovers per 300 kb per F$_2$ plotted along the Arabidopsis chromosome 1. Mean values are shown by horizontal dashed lines. SNPs per 300 kb are plotted and shaded in grey. The position of centromere is indicated as vertical dashed line. **e** Data as for (**d**), but analysing crossovers along proportionally scaled chromosome arms, orientated from telomere (TEL) to centromere (CEN). **b**–**e** Data for wild type, *msh2* and *recq4* from refs. 41,49,66, respectively. Source data are provided as a Source Data file.

however, is much stronger in regions of relatively low polymorphism densities.

### Local effects of DNA polymorphism on Class II crossovers

In our previous studies, we observed that heterozygous regions show elevated crossover numbers when they are adjacent to homozygous regions on the same chromosome[48]. As inferred from genetic analysis, this effect depends on MSH2 detecting interhomolog polymorphisms[49]. Interestingly, the hetero-/homozygosity juxtaposition effect applies only to Class I crossovers, as in *fancm zip4* double mutants, a strong decrease in crossover frequency in the heterozygous region was observed[48]. To investigate the genetic basis of this distinct crossover response to interhomolog polymorphism, we measured crossover frequency in plants with different patterns of heterozygosity. For this purpose, we used Col/Ct recombinant lines (where Ct stands for the *A. thaliana* accession, Catania), which differ in the heterozygosity pattern along chromosome 3. We measured crossover frequency in the *420* interval, which is also located on chromosome 3[48].

Four Col/Ct heterozygosity combinations were analysed: (i) "HOM*420*-HOM" that are Col/Col homozygous throughout the genome, (ii) "HET*420*-HET" that are Col/Ct heterozygous throughout the genome, (iii) "HET*420*-HOM" where the *420* region is Col/Ct heterozygous and the remainder of chromosome 3 is Col/Col homozygous and (iv) "HOM*420*-HET" where *420* is Col/Col homozygous and the remainder of chromosome 3 is Col/Ct heterozygous (Fig. 4a). The *420* index used in the line name indicates the location of the *420* interval in the homozygous (HOM) or heterozygous (HET) region.

We observed a very high recombination frequency (-30–40 cM) in the *fancm zip4* mutant whenever the *420* region was homozygous, and low (-11–18 cM) whenever *420* was heterozygous (Fig. 4b, c), which is consistent with previous observations[48]. When we additionally inactivated the *MSH2* gene in these lines, obtaining the triple mutants *msh2 fancm zip4*, *420* crossover frequency for HET*420*-HOM (24.0 cM) and HET*420*-HET (27.8 cM) significantly increases compared to their counterparts in the *fancm zip4* background (Welch's *t* test $P = 1.1 \times 10^{-13}$ and $P < 2.2 \times 10^{-16}$, respectively), while crossover frequency was unchanged for the HOM*420*-HET and HOM*420*-HOM lines. This observation shows that Class II crossovers are repressed by MSH2 in the heterozygous (polymorphic) regions both in full hybrids (HET*420*-HET) and in lines with an alternating pattern of polymorphism (HET*420*-HOM) (Fig. 4c).

The increase in *420* crossover frequency in *msh2 fancm zip4* HET*420*-HET is also significant compared to its wild-type HET*420*-HET counterpart (Welch's *t* test $P < 2.2 \times 10^{-16}$) (Fig. 4b, c). However, both the HET*420*-HET and HET*420*-HOM lines in *msh2 fancm zip4* remain lower recombining in *420* than HOM*420*-HOM and HOM*420*-HET in *msh2 fancm zip4*, indicating that genetic or epigenetic factors remains that limit the formation of crossovers in the heterozygous state (Fig. 4c). This is consistent with a MSH2-independent effect of DNA polymorphism on Class II crossover formation.

We created combinations carrying only *fancm* or *msh2 fancm* mutations, thus having both active Class I and increased Class II crossovers (Fig. 4d). As expected, *420* crossover frequency for the juxtaposition lines in *fancm* is the intermediate of frequencies observed in wild type and *fancm zip4* (Fig. 4d). Similarly, the values measured for *msh2 fancm* appear to be the intermediate of the crossover frequencies measured for *msh2* and *msh2 fancm zip4* (Fig. 4d). This result confirms that MSH2 has opposite effects on Class I and Class II crossovers in response to interhomolog polymorphism.

Previously, we showed that overexpression of *HEI10* preserved the hetero-/homozygosity juxtaposition effect (Fig. 4e)[49]. When we combined *HEI10* overexpression with *MSH2* inactivation, this effect disappeared. Instead, *msh2 HEI10-OE* showed *420* crossover frequencies for particular Col/Ct heterozygosity combinations similar to the ones observed in the single *msh2* mutant, though all the lines showed increases in crossovers (compare Fig. 4b and e). In addition, *HEI10* overexpression was not able to increase crossover frequency above the level observed in *msh2 fancm zip4* triple mutants (Fig. 4c). Altogether these results confirm that HEI10 has no role in crossover formation outside the Class I pathway.

### The pericentromere of chromosome 3 shows MSH2-independent crossover inhibition when heterozygous

We decided to test if *msh2* is able to increase *fancm zip4* crossover frequency in the polymorphic pericentromeric regions. To this end, we backcrossed *msh2*, *fancm zip4* and *msh2 fancm zip4* mutations to the CTL-*3.9* FTL[67] (hereafter *3.9*), in which the marked interval spans the centromere and pericentromere of chromosome 3. We obtained HOM-HOM*3.9* (Col homozygous throughout the genome), HET-HET*3.9* (Col/Ct heterozygous throughout the genome) and HET-HOM*3.9* (Col/Ct heterozygous in the first 5 Mb of the chromosome 3, but Col homozygous throughout the remainder of the chromosome, including the *3.9* interval), lines in mutant and wild-type backgrounds. The *3.9* index in the line name indicates the location of the *3.9* interval in the HOM or HET regions. The location of the *3.9* interval with respect to the pattern of heterozygosity in these lines is shown on Fig. 4a.

In wild-type *3.9* lines, crossovers were higher in HET-HET*3.9* than in HOM-HOM*3.9* (20.24 and 17.44 cM, respectively; Welch's *t* test $P = 1.5 \times 10^{-3}$; Fig. 4f), which is consistent with previous observations that Col/Col inbreds have a relatively low crossover frequency in the pericentromeric regions[40]. HET-HOM*3.9* lines, where the *3.9* interval is in the homozygous region, while the *420* interval is heterozygous, showed a decrease of *3.9* crossover frequency compared to HOM-HOM*3.9* (15.74 cM, $P = 9.6 \times 10^{-3}$; Fig. 4f). This decrease is due to crossover redistribution from the homozygous region spanning *3.9* interval, to the heterozygous sub-telomeric region, which is consistent with the hetero-/homozygosity juxtaposition effect[48].

In the *msh2* background, we observed that *3.9* crossover frequency is significantly lower in HET-HET*3.9* than in HOM-HOM*3.9* backgrounds (14.30 and 18.47 cM, $P = 2.8 \times 10^{-4}$), whereas HOM-HOM*3.9* was not different from HET-HOM*3.9* (Fig. 4f). This shows that when MSH2 is inactive, crossover recombination is inhibited within the pericentromeres when they are heterozygous. One possibility is that this is triggered by a high number of mismatches formed during interhomolog strand invasion, because the pericentromeres are substantially more polymorphic than distal regions[68]. Alternatively, differences in chromatin states between Col and Ct accessions within pericentromeres may be responsible[69].

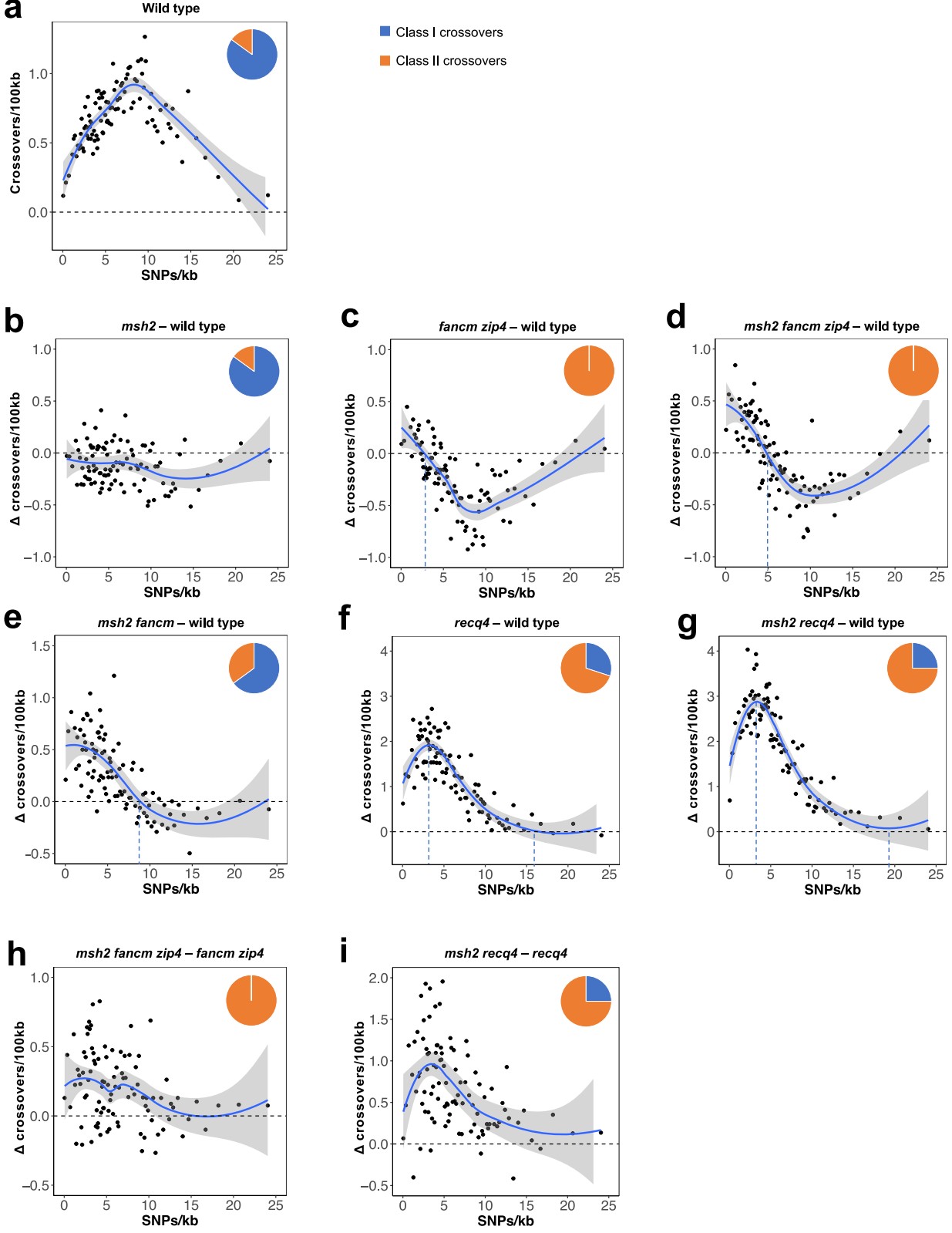

**Fig. 3 | Relationship between SNP density and crossover frequency in different mutant backgrounds. a** Crossover frequency as a function of SNP density (SNPs/kb) in wild-type Col/*Ler* plants. Crossovers normalised by the number of F₂ individuals and SNP density in 100 kilobase (kb) adjacent windows were calculated for each population and ranked into percentiles according to SNP density. **b**–**g** The difference between crossover frequency in a mutant and wild type (Δ cM) was plotted in relationship to SNP density (SNPs/kb). **h**–**i** The difference between crossover frequency in a multiple mutant carrying *msh2* mutation and its counterpart with functional *MSH2* (Δ cM) was plotted in relationship to SNP density (SNPs/kb). Trend lines were fitted in ggplot2 using Local Polynomial Regression Fitting (loess) with the formula y ~ x. The ratio between Class I (blue) and Class II (orange) crossovers, estimated based on genome-wide crossover mapping for each background is printed inset. Data for wild type, *msh2* and *recq4* from refs. [41],[49],[66], respectively. Source data are provided as a Source Data file.

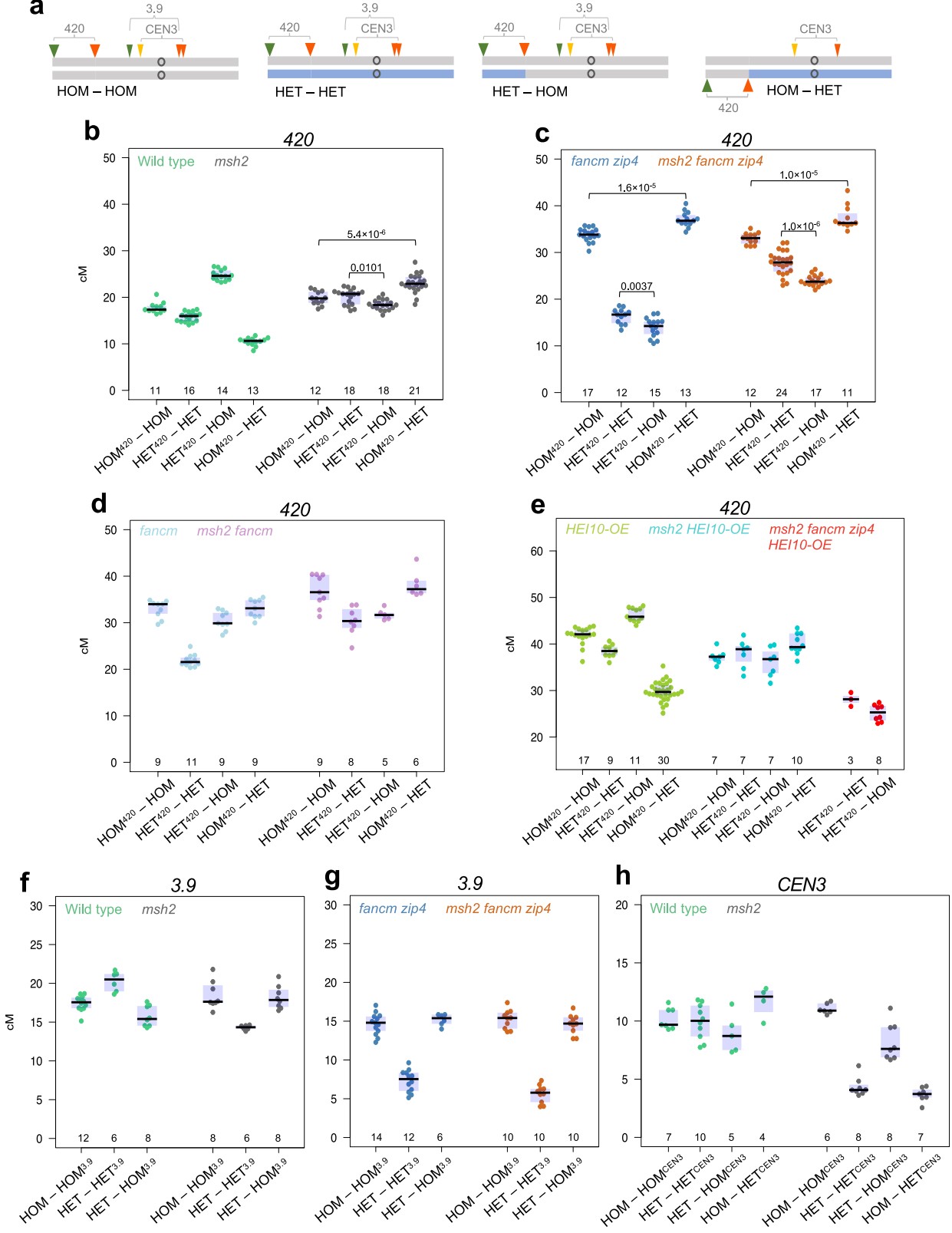

For any of the heterozygosity combinations in the *fancm zip4* double and *msh2 fancm zip4* triple mutants tested, we did not observe a significant increase in *3.9* crossovers relative to wild type (Fig. 4f, g). In both *fancm zip4* and *msh2 fancm zip4*, *3.9* crossovers in the lines differing in the pattern of heterozygosity is very similar to that observed in the *msh2* mutant (compare with Fig. 4f), but the decrease in HET-HET³·⁹ relative to HOM-HOM³·⁹ and HET-HOM³·⁹ is stronger. This

confirms previous observations that a mutation in the *FANCM* gene is unable to restore crossovers in the heterozygous pericentromeric regions when the Class I pathway is mutated[48]. However, it also reveals that this inability of Class II crossovers to be formed in polymorphic regions in *fancm* mutants is mainly MSH2-independent. These results are consistent with our genome-wide data for the Col/L*er* hybrids, which also show relatively low crossovers in pericentromeres (Fig. 2e).

**Fig. 4 | MSH2-dependent and independent crossover redistribution in response to heterozygosity pattern. a** Ideograms of chromosome 3 in lines differing in heterozygosity pattern. Grey corresponds to Col while blue corresponds to Ct genotype. Location of fluorescent reporters defining three different intervals (*420*, *3.9* and *CEN3*) are indicated together, for simplicity. **b** *420* crossover frequency (cM) in the HOM-HOM, HET-HET, HET-HOM, HOM-HET genotypes shown in (**a**), in either wild type or *msh2*. Underlined is the homo- or heterozygosity state of the *420* interval. The centre line of a boxplot indicates the mean; the upper and lower bounds indicate the 75th and 25th percentiles, respectively. Each dot represents a measurement from one individual. The numbers of individuals are also indicated below the boxplots. The two-sided *P* values were estimated by Welch's *t* test. **c** As in (**b**), but for *fancm zip4* and *msh2 fancm zip4*. **d** As in (**b**), but for *fancm* and *msh2*

*fancm*. **e** As in (**b**), but for *HEI10-OE*, *msh2 HEI10-OE* and *msh2 fancm zip4 HEI10-OE* (only HET-HET and HET-HOM genotypes). **f** *3.9* crossover frequency (cM) in the HOM-HOM, HET-HET and HET-HOM genotypes shown in **a**, in either wild type or *msh2*. A boxplot is defined as in (**b**). **g** As in (**f**), but for *fancm zip4* and *msh2 fancm zip4*. Underlined is the homo- or heterozygosity state of the *3.9* interval. **h** *CEN3* crossover frequency (cM) in the HOM-HOM, HET-HET, HET-HOM, HOM-HET genotypes shown in (**a**), in either wild type or *msh2*. A boxplot is defined as in (**b**). Underlined is the homo- or heterozygosity state of the *CEN3* interval. Each dot represents a measurement from a pool of 5–8 individuals. The numbers of pools are indicated below the boxplots. Data in (**b**) from ref. 49 Source data are provided as a Source Data file.

We confirmed these results for *msh2* using the *CEN3* FTL interval, which also has a pericentromeric location and partially overlaps with *CTL-3.9* (Fig. 4a). In the case of *CEN3*, recombination is measured by segregation of fluorescent reporters expressed in the pollen, therefore it is specific for male meiosis[63,70]. These results are consistent with those observed for *3.9*, except that the decrease in *msh2* HET-HET$^{CEN3}$ recombination relative to *msh2* HOM-HOM$^{CEN3}$ is stronger (compare *msh2* data on Fig. 4f and h; 4.3 and 11.06 cM, respectively $P = 4.6 \times 10^{-10}$). This may be related to the more distalized crossover frequencies observed in Arabidopsis male meiosis than in female meiosis (i.e., crossover frequency is higher in sub-telomeric regions in male meiosis compared with female)[71,72].

Altogether these results show that MSH2 is effective at stimulating Class I crossovers in the pericentromeric regions when they are heterozygous. In contrast, non-interfering Class II crossovers that form in the absence of FANCM cannot be formed in the polymorphic regions near the centromere, whether MSH2 is active or not.

### Crossover redistribution along the chromosome arms in response to the patterns of heterozygosity

One of the advantages of using reporter systems for measuring crossover frequency is that they allow us to study recombination in both hybrid and otherwise inbred contexts. We decided to use FTLs to investigate whether interhomolog polymorphism-dependent crossover redistribution (i.e., the heterozygosity/homozygosity juxtaposition effect[48]) is uniform along heterozygous regions. For this purpose, we used a set of eight additional FTLs that cover the long (left) arm of chromosome 3 (Fig. 5a and Supplementary Table 2)[67]. We previously generated lines with HET-HET, HOM-HOM and HET-HOM homo-/heterozygosity combinations along this chromosome arm using Col/Ct accessions (Fig. 5a)[48,49].

We first compared the distribution of crossovers along the arm in hybrids (HET-HET) versus inbreds (HOM-HOM) (Fig. 5b and Supplementary Fig. 6). We observed a slight though significant increase in HET-HET crossover frequency in the centromere-proximal regions with a concomitant reduction in the arm (Fig. 5b). This change corresponds to the effects recently described in a genome-wide comparison of Col/Col quasi-homozygous lines (with a low number of markers introduced by mutagenesis) with Col/Ler hybrids[40]. In the HET-HOM line, where the distal regions of the chromosome was heterozygous, while the rest of the chromosome was homozygous, there was a strong increase in the crossover frequency in the HET region compared to both HOM-HOM and HET-HET, but only at the HET-HOM border (intervals D and partly E, which is ~2 Mb from the HOM/HET border; Fig. 5c, d and Supplementary Fig. 6). As expected, this crossover increase was at the expense of a decrease in the neighbouring HOM region (intervals F to H on Fig. 5c, d). Interestingly, a statistically significant decrease was observed not only in the immediate vicinity of the HET/HOM border, but over a longer region of the chromosome, including the "H' pericentromeric interval, which starts 3.47 Mb from the HET-HOM breakpoint (Fig. 5c, d). These results indicate that the increase in crossover frequency in the juxtaposition effect occurs only

locally, close to the border between the homozygous and heterozygous regions.

To investigate how MSH2 influences crossover distributions in different heterozygosity contexts, we backcrossed the eight FTLs to the *msh2* knock-out mutant. Comparisons between *msh2* HOM-HOM (inbreds) and (HET-HET) hybrids revealed that only the pericentromeric interval 'H' showed significant differences in crossover frequency ($P = 9.5 \times 10^{-5}$; Fig. 5e and Supplementary Fig. 6). This may be due to potential structural variations between Col and Ct within the pericentromeres. Crossover distribution does not differ between *msh2* HET-HOM and HET-HET outside this region (Fig. 5f). However, we observed a significant increase in crossover frequency in HET-HOM versus HOM-HOM in the 'F' interval in *msh2* ($P = 8.3 \times 10^{-4}$; Fig. 5g), which is opposite to wild type ($P = 1.7 \times 10^{-5}$; Fig. 5d). Overall, our data show that the differences in crossover distributions between lines differing in status and pattern of heterozygosity are much smaller in *msh2* than in wild type.

In conclusion, our analyses confirm that the chromosomal crossover distribution is similar in hybrids and inbreds, as recently suggested by genome-wide analysis[40]. Only the introduction of a heterozygous chromosomal segment into an otherwise homozygous chromosome causes a drastic local redistribution of crossover recombination, and this effect is MSH2-dependent. This demonstrates that MSH2 has a stimulating effect on Class I crossovers in response to interhomolog polymorphism, primarily in the immediate boundary between heterozygous and homozygous regions. However, in the case of a gradual decrease in polymorphism density, which exists in hybrids, the effect is weak.

### Crossover interference is maintained both in *msh2* hybrids and inbreds

Linked fluorescent reporters expressed in pollen can be used to locally measure crossover interference in adjacent intervals represented as coefficient of coexistence $(1 - CoC)$[63,73]. This provides an opportunity to compare the strength of interference both in hybrids and inbreds, either in wild-type or *msh2* backgrounds (Supplementary Fig. 7). Consistent with previous reports[48], we observed that interference is stronger in hybrids than in inbreds (Supplementary Fig. 7c). In contrast, interference remained unchanged in both *msh2* inbreds and hybrids relative to the values measured for wild type (Supplementary Fig. 7c). This shows that although the presence of interhomolog polymorphism increases the strength of crossover interference, this effect is not dependent on the detection of mismatches involving MSH2.

### Discussion

In wild-type *A. thaliana*, the ZMM pathway that creates Class I crossover is dominant, while Class II crossovers are rare and therefore insufficient to secure balanced segregation of chromosomes in meiosis[74,75]. To investigate the differences between the effect of interhomolog polymorphism on both crossover classes, we used mutants in which Class II is increased (*fancm* and *recq4*), and in

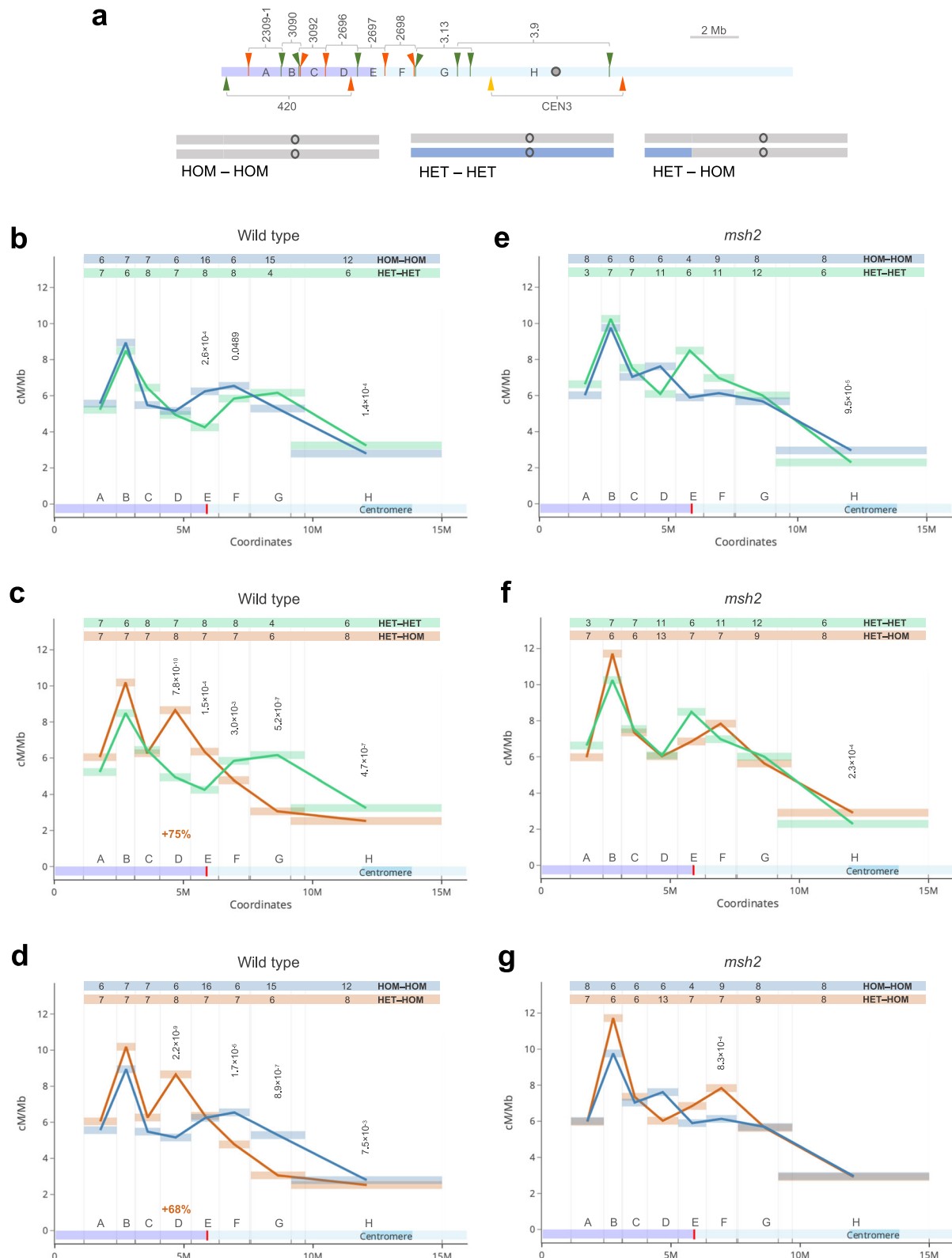

conjunction with the *zip4* mutation that blocks Class I crossovers[20,23,24]. It should be noted, however, that it is not entirely certain whether Class II crossovers formed in the *fancm* and *recq4* mutants behave in the same way as wild-type Class II crossovers.

Previous studies have shown that while combination of *fancm* and *zip4* mutations leads to normal fertility in inbred backgrounds, repeating this experiment in hybrids results in sterile plants[24]. An explanation for this is that interhomolog polymorphism blocks Class II crossovers[6,24]. Indeed, the combination of the *fancm zip4* mutation with the *msh2* mutation resulted in an increase in fertility, but not to the level observed in wild type (Fig. 1). Therefore, we performed genome-wide crossover analysis for *fancm zip4*, *msh2 fancm zip4*, *recq4* and *msh2 recq4* Col/Ler hybrids. This revealed that *msh2* mutations invariably lead to increases in Class II crossover numbers (Fig. 2),

**Fig. 5 | Crossover distribution across the chromosome arm in response to polymorphism. a** (upper panel) Location of eight FTL intervals across the Arabidopsis chromosome 3 shown in addition to *420* and *CEN3* intervals. Fluorescent reporters are indicated by tick marks and arrowheads with colours corresponding to eGFP (green), dsRed (red) and eYFP (yellow). Capital letters 'A' – 'H' were used instead of the original FTL names (indicated on the top), for simplicity. The violet and light blue shading represents 'HET' and 'HOM' regions in HET-HOM line, respectively. Genetically defined centromere[95] is indicated as grey circle. (lower panel) Ideograms of chromosome 3 showing heterozygosity pattern in lines used in (**b**–**g**). Grey corresponds to Col while blue corresponds to Ct genotype.

**b**–**d** Comparison of mean crossover frequency (cM/Mb) in eight intervals along the chromosome arm in inbred (HOM-HOM) vs. hybrid (HET-HET) (**b**), hybrid vs. recombinant line (HET-HOM) (**c**) and inbred vs. recombinant line (**d**). Rectangles represent the length of the intervals, interval names as in (**a**). The numbers of individuals used to calculate the means are indicated at the top and highlighted in the colour corresponding to the given genotype. One-way ANOVA with Tukey HSD was used to calculate statistical significance. Not significant values were not shown. The relative percentage increase in the interval D crossover frequency is indicated in red. **e**–**g** As in (**b**–**d**), but for the *msh2* mutant. Source data are provided as a Source Data file.

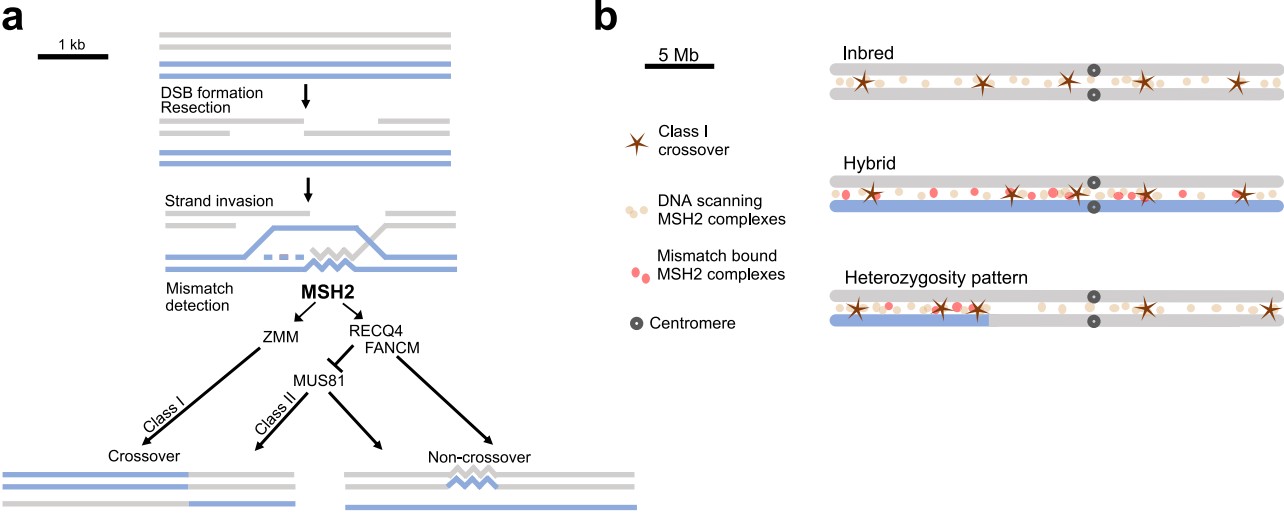

**Fig. 6 | Models showing the impact of DNA polymorphism on crossover formation during Arabidopsis meiosis. a** Two recombining homologous DNA molecules are depicted in grey and blue, over a region of several kilobases. Following DSB formation on the grey molecule, resection occurs to form 3'-single-stranded DNA. This ssDNA undergoes strand invasion into the homologous molecule, forming a displacement loop. MSH2 heterodimers detect mismatches at the invasion site. Two scenarios are proposed for the situation when mismatches are detected: (i) MSH2 promotes ZMM pathway leading to Class I crossover, or (ii) MSH2 recruits or stimulates DNA helicases, including FANCM and RECQ4, resulting in D-loop displacement and non-crossover repair. In the absence of mismatches or MSH2, MUS81 endonuclease repairs the DSB via Class II crossover or non-crossover.

Alternatively, MSH2 can directly stimulate MUS81-dependent crossover formation (not shown). **b** Two recombining homologues are depicted with grey and blue colours representing sequence divergence. MSH2 complexes scan DNA to detect mismatches in heteroduplexes. In inbreds, when there are no mismatches, the position of crossovers is determined mainly by the chromatin structure. In hybrids, the mismatches along the entire chromosome length trigger a fairly even distribution of mismatch-bound MSH2 complexes, which, combined with interference, also results in Class I crossover placement determined mainly by chromatin. However, the presence of a single heterozygous region on an otherwise homozygous chromosome results in a local concentration of mismatch-bound MSH2 complexes that stimulate Class I crossover in the heterozygous region.

indicating that MSH2, presumably via binding mismatches during interhomolog strand invasion, blocks Class II crossover either by recruiting FANCM and RECQ4, or by limiting MUS81 activity (Fig. 6a). In budding yeast, MSH2 complexes recruit SGS1, the RECQ4 homologue, to heteroduplexes, thus prevents crossover repair in somatic cells[76,77]. Interestingly, in both *fancm* and *recq4*, the *msh2* mutation is unable to force Class II crossovers in regions with high polymorphism density (Fig. 3). This suggests that DNA polymorphism also inhibits Class II crossovers in an MSH2-independent manner, possibly limiting the stability of D-loops during strand invasion.

In addition, we investigated how MSH2 inactivation affects the distribution of Class II crossovers when the same chromosome fragment is either homozygous or heterozygous. This allows the elimination of other effects besides polymorphism, including DNA methylation and chromatin state. We confirmed that FANCM-associated Class II crossovers are strongly repressed when the region is heterozygous and *msh2* mutation only partially restores recombination activity (Fig. 4). At the centromere-proximal region in a heterozygous state, *msh2* is unable to increase Class II crossover frequency, remaining twofold lower than when this region is homozygous (Fig. 4f–h). Again, this confirms our observation that polymorphism inhibits Class II crossovers in a MSH2-independent manner.

Although we do not have mutants in which only the ZMM pathway is active, the Class I crossovers predominate both in wild type and in *msh2*[2,3,8,28]. Our genome-wide analyses showed a limited effect of polymorphism on Class I crossover distributions (Figs. 2 and 3, ref. 49), which is consistent with previous observations[40]. Using reporter lines covering the entire left arm of chromosome 3 we showed that the differences between inbreds and hybrids are small (Fig. 5). However, the juxtaposition of the heterozygous and homozygous regions causes a MSH2-dependent redistribution of the Class I crossover towards the former, regardless of the chromosomal location[48,49] (Figs. 4b, f, h and 5). Recently, we showed that this applies not only to large chromosomal regions, but also to individual recombination hotspots (i.e., regions of a few kilobases in size showing elevated crossover frequency)[50]. This is consistent with a positive role of MSH2 on Class I crossover in response to interhomolog polymorphism primarily in the situation of a local change of polymorphism density along the chromosome. In line with this, in vitro assays show that yeast MSH2 complexes stimulate MLH1-MLH3, the major endonuclease in the ZMM pathway[78].

However, Arabidopsis hybrids naturally have regions of higher and lower interhomolog polymorphism density (Fig. 2d and Supplementary Fig. 4). This raises the question of why we don't observe

significant crossover redistribution between inbreds and hybrids? A possible explanation may be the limited availability of MSH2 heterodimers. In hybrids, the SNP density may be high enough that not all mismatches can be bound simultaneously by MSH2 complexes. As crossover interference remains at a similar level in wild type and *msh2*, tight crossover control is maintained (Supplementary Fig. 7, ref. 49). As a consequence, other factors, such as chromatin structure or DNA methylation, may become dominant, while polymorphism-dependent changes in crossover distribution are relatively small (Fig. 5b, ref. 40). On the contrary, when only a single chromosomal region is heterozygous and the rest of the genome is homozygous, MSH2 saturation occurs in this heterozygous region triggering a strong local crossover stimulation (Fig. 6b). Similarly, in a homozygous region on an otherwise heterozygous chromosome, the absence of MSH2 results in a decrease in crossovers. This hypothesis is confirmed by the fact that we do not observe many differences in crossover distribution between the tested lines differing in patterns of heterozygosity in *msh2* (Fig. 5e–g).

Our results show that the two crossover pathways exhibit dramatic differences in response to interhomolog polymorphisms and that the effect of MSH2 complexes on these pathways is opposite (Fig. 6a). The observed differences are likely related to different biological functions of the two pathways. The ZMM pathway leads to the formation of Class I crossover and is dedicated exclusively to meiotic recombination during gamete formation. As sexual reproduction involves mixing genetic material from non-identical parental individuals, the detection of polymorphisms in this pathway cannot block crossovers. Moreover, biasing crossovers to regions that differ between individuals allows for the formation of new allelic combinations. This is different for Class II crossovers, which are formed via pathways that are shared with DNA repair in somatic cells, where recombination between non-identical sequences threatens genome stability[79,80]. Therefore, the detection of mismatches blocks Class II crossover repair, leading to heteroduplex rejection and non-crossover repair.

A lack of meiotic recombination is one of the causes of infertility in divergent hybrids[81,82]. Our results indicate that the inactivation of the mismatch detection system, in combination with the inactivation of anti-recombination factors, enables an increase in the frequency of Class II crossover in polymorphic regions. A similar effect was recently reported in hybrids between diverged *Saccharomyces* species[83]. The use of such an approach may allow breakage of reproductive isolation between plant species, where the limitation is DNA divergence, which prevents the exchange of genetic material.

## Methods

### Growth conditions and plant material
Plants were grown in controlled environment chambers at 21 °C with long day 16/8 h light/dark photoperiods with 70% humidity and 150-µmol light intensity. Prior to germination seeds were kept for 48 h in the dark at 4 °C to stratify germination.

The *msh2-1* T-DNA insertion line (SALK_002708) and Arabidopsis accessions Col, L*er* and Ct were obtained from the Nottingham Arabidopsis Stock Centre (NASC). The *msh2-2, msh2-3, msh2-4, msh2-5* (in Col/Ct recombinant lines) and *msh2-6* (in L*er*−0) deletion mutants were previously generated in our laboratory via CRISPR/Cas9 mutagenesis[49,50], as well as L*er zip4-3* mutant generated in this study. The *HEI10-OE* line corresponds to transgenic line "C2", previously described in ref. 16. For measuring recombination frequency fluorescence-tagged lines were used: *420* (kindly provided by Avraham Levy[62]), *Cen3* and *I3bc* (kindly provided by Gregory Copenhaver[63]), CTLs *3.9, 3.13, 2309-1, 3090, 3092, 2696, 2697, 2698* (kindly provided by Scott Poethig[67]). The Col mutants *fancm-1, zip4-2, recq4a-4, recq4b-2* and the L*er* mutants *recq4a* and *fancm-10* were kindly provided by Raphael Mercier[20,23,57,84]. All primer sequences

used for genotyping of mutant lines are described in Supplementary Table 1.

### Genotyping-by-sequencing library preparation
DNA was extracted from leaves of F$_2$ plants obtained from Col × L*er* cross. DNA extraction was performed as described[64] and the quality of the DNA was checked in 1% agarose gel. Tagmentation was performed by mixing 1 µL of 5 ng/ µL of the DNA with 1 µL of Tagmentation Buffer (40 mM Tris-HCl pH=7.5, 40 mM MgCl2), 0.5 µL of DMF (Sigma), 2.35 µL of Nuclease-free water (Thermo Fisher) and 0.05 µL of loaded, in-house produced Tn5. Loading Tn5 with the annealed linker oligonucleotides was previously described[85]. The tagmentation step was carried out at 55 °C for 2 min and then stopped by adding 1 µL 0.1% SDS and incubating at 65 °C for 10 min. Amplification of the tagmented DNA was performed using the KAPA2G Robust PCR kit (Sigma) and custom P5 and P7 indexing primers. Each sample was amplified with the unique set of P5 and P7 primers as described[41]. The successful libraries were pooled and size selected in 2% agarose gel, after which DNA fragments in a range of 400–700 bp were excised and extracted using Gel Extraction Kit (A&A Biotechnology) The quality and quantity of the libraries were verified with TapeStation system (Agilent) and Qubit 2.0 fluorometer. Paired-end sequencing of libraries was performed on HiSeq X-10 instrument (Illumina).

### Genotyping-by-sequencing (GBS) bioinformatics analysis
To identify SNPs within *fancm zip4, msh2 fancm, msh2 fancm zip4* and *msh2 recq4* Col×Ler F$_2$ populations, demultiplexed paired-end forward and reverse reads have been pooled and aligned to Col-0 genome reference sequence with use of BowTie2[86]. Resulting BAM files have been sorted and indexed with use of SAMtools v1.2[87]. SNPs were called using SAMtools and BCFtools[88]. Subsequently, individual sequencing libraries have been aligned to Col-0 genome reference sequence with default parameters in BowTie2 and compared to the previously generated SNP list with SAMtools and BCFtools. Later, the resulting tables of SNPs have been filtered to keep only SNPs with high mapping quality (>100) and high coverage (> 2.5×) in R. Individual libraries with less than 100,000 reads were discarded from the analysis. To call crossovers TIGER pipeline has been used on filtered files[64]. A summary of GBS results is presented in Supplementary Table 3. To investigate crossover distribution, crossover frequencies have been binned into scaled windows and summed across chromosome arms.

For analysis of the relationship between crossover recombination and SNP density, the genome was divided into 100 kb non-overlapping windows and for each of them SNP density was determined based on published Col/L*er* polymorphism data[89]. The crossover frequency per each window was normalised to the number of analysed individuals. This resulted in 1191 windows, which were sorted according to the SNP density and grouped into 99 groups, so that each group consisted of 12 windows with a similar polymorphism level (Supplementary Fig. 5). For each SNP density group crossover number were calculated based on GBS data and normalised to the sample size. The crossover number, or a difference between crossover number for different backgrounds were plotted for each SNP density group against SNP density. Trend lines were fitted in ggplot2 using Local Polynomial Regression Fitting (loess) with the formula y ~ x.

### Crossover frequency measurement using FTL seed-based system
Crossover rate measurements using seed-based system were performed as described previously[48,90]. Briefly, pictures of seeds were acquired using epifluorescent microscope in bright field, ultraviolet (UV) + dsRed filter, and UV + GFP filter. The images were later processed by CellProfiler software[91] to identify seed boundaries and to assign a dsRed and eGFP fluorescence intensity value to each seed object. Thresholds between fluorescent and non-fluorescent seed were

set manually using fluorescence histograms for each colour. The crossover frequency is calculated as cM = 100 × (1−(1 − 2(NG + NR)/NT)/2), where NG is the number of green alone seeds, NR is the number of red alone seeds, and NT is the total number of seeds.

## Crossover frequency and interference measurements using FTL pollen-based system

Samples for flow cytometry analysis were prepared as described previously[73]. Inflorescences from 5-8 individual plants were pooled for each experimental variant, with at least three biological replicates. The flow cytometry was performed on Guava easyCyte 8HT Cytometer (Millipore). The samples were analysed using GuavaSoft 3.3 programme (Millipore). Obtained events were separated based on forward and side scatter and hydrated pollen was gated to exclude dead or damaged material. For crossover frequency measurements in *CEN3* interval the events were gated into four classes based on their fluorescence emission signals: red (R), yellow (Y), double-colour (RY) and non-colour (N). Crossover frequency (cM) was calculated as 100 × (Y/(Y + RY)). For *I3bc* interval crossover interference measurements events were divided into eight classes. *I3b* and *I3c* genetic distances were calculated by dividing the sum of recombinant gametes in particular interval by the total number of pollen grains. Crossover interference was calculated by counting the coefficient of coincidence (CoC), which is the ratio between the expected and the observed double crossover (DCO) number. The expected DCO frequency is obtained by dividing by hundredths the genetic distances in *I3b* and *I3c* intervals and further multiplying them by the total number of pollen grains. The observed DCO is the sum of pollen grains that have experienced a double crossover (B-R and -Y- classes). Interference is then calculated as follows: 1 − CoC.

## CRISPR/Cas9 mutagenesis of *ZIP4* in L*er*−0

CRISPR/Cas9 mutagenesis on Arabidopsis plants was performed according to the protocol[92,93]. To obtain *zip4* mutant line in L*er*−0 background, a pair of gRNAs targeted within exon 1 of *ZIP4* were designed. A vector containing the *ZIP4* gRNA pair under the U3 and U6 promoters, and a *ICU2::Cas9* transgene was used for *Agrobacterium* transformation. Transformants were genotyped by PCR amplification with primers flanking the *ZIP4* gRNA target sites (Supplementary Fig. 1a, b). Sanger sequencing was performed to detect deletions−mutants with heritable deletions causing a frame shift in *ZIP4*, and not carrying the CRISPR-Cas9 construct, were identified for further experiments.

## Fertility assays

Seed set and silique length were assessed from five fruits, located at positions 6 through 10 of the main stem, in eight plants per genotype. Collected siliques were incubated in 70% EtOH for 72 h and later photographed in the bright field using bottom light source, enabling seed set calculations, which were performed manually. Silique length was calculated using ImageJ software[94].

## Chiasmata counting

Genotypes were grown together, and primary inflorescences were collected at the same time from three individual plants for each genotype. Metaphase I chromosome spread preparations were prepared from ethanol: acetic acid (3:1) fixed material. 20 cells were imaged and analysed per each individual. Chiasma counts were based on the shape of bivalents.

## Statistics and reproducibility

All experiments were conducted under the same standard conditions (see Growth conditions). Three individuals per genotype were used to count chiasmata, eight plants per genotype were used for fertility assays, and at least six biological replicates were used for crossover

frequency measurements in FTLs, with the exception of interval A (2309-1) in the *msh2* background in HET-HET, where only three individuals were used, and interval E (2697) in the *msh2* background, where four individuals were used (Supplementary Fig. 6). The number of replicates was dictated solely by the availability of material. No data were excluded from the analyses. For each biological replicate, the crossover frequency was measured based on the segregation of ~1500 to ~2500 seeds (seed-based system) and from ~8000 to ~40,000 pollen grains (pollen-based system). The sample size resulted from the number of seeds produced by an individual plant or the amount of pollen extracted, and is similar to analogous analyses published previously.

For genome-wide crossover distribution, the analysis was performed based on one randomly selected F[1] individual per each genotype, for which 175 to 279 F[2] individuals were sequenced. This sample size was estimated based on previously published studies were the effect of F[2] population size on the fidelity of reproducing the crossover distribution was evaluated[41,49,64]. Up to ten F[2] samples per genotype were eliminated from the final analyses due to the low number of sequencing reads (less than 100,000 reads).

## Reporting summary

Further information on research design is available in the Nature Portfolio Reporting Summary linked to this article.

## Data availability

All data generated for this study are included in the published version of the article, or its supplementary data files. The GBS sequence data generated in this study (for *msh2 fancm*, *msh2 fancm zip4*, *msh2 recq4ab* and *fancm zip4*) have been deposited in the NCBI Sequence Read Archive (SRA) under the BioProject accession codes PRJNA952840. Raw GBS data for the wild-type Col×L*er* F[2] population were downloaded from ArrayExpress E-MTAB-8165[41]. Raw GBS data for *recq4* Col×L*er* F[2] were downloaded from ArrayExpress E-MTAB-5949[66]. Raw GBS data for *msh2* Col×L*er* F[2] were downloaded from ArrayExpress E-MTAB-8252[49]. The Col-0 TAIR10 reference genome is downloaded from the TAIR database. The sequence polymorphism data for the Col/L*er* cross used in this study was downloaded from https://1001genomes.org/projects/MPIPZJiao2020/index.html. Seed scoring and pollen scoring raw data generated in this study are provided in the Source data file. Source data are provided with this paper.

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

## Acknowledgements

We thank Raphael Mercier (Max Planck Institute for Plant Breeding Research, Cologne) for sharing *fancm zip4*, *recq4a recq4b* mutants in

Col and L*er* backgrounds. We thank Michal R. Gdula (Adam Mickiewicz University, Poznan) for the critical reading of the manuscript. The computations were performed at the Poznan Supercomputing and Networking Center (grant 312). This work was supported by the National Science Center, Poland (NCN) grants 2016/22/E/NZ2/00455 and 2020/39/I/NZ2/02464 to P.A.Z., 2021/41/N/NZ2/01226 to J.D., the Foundation for Polish Science grant (POIR.04.04.00-00-5C0F/17-00) to P.A.Z.

## Author contributions

J.D. and P.A.Z. designed research; J.D., M.Sz.-L., M.G. and J.H. performed research; I.R.H. contributed computational pipelines and materials; W.D. performed the computational analyses; J.D., W.D., M.Sz.-L., J.H. and P.A.Z. analysed the data; and J.D. and P.A.Z. wrote the paper with the aid of all authors.

## Competing interests

The authors declare no competing interests.
