## [Peer Review File · Nature Communications]

MSH2 stimulates interfering and inhibits non-interfering crossovers in response to genetic polymorphismREVIEWER COMMENTS

Reviewer #1 (Remarks to the Author):

OVERVIEW

Recombination is critical to generate genetic diversity within sexually reproducing organisms, but how it is controlled and influenced by features such as DNA sequence polymorphisms that exist within sexually breeding species is unclear. The authors use *Arabidopsis* to explore the effect that the key mismatch repair protein, MSH2 has on recombination in meiosis—specifically in crossover formation—in various mutants that alter the relative proportion of class I and class II COs. The study is detailed and extensive, however, I found many statements and inferences to lack sufficient explanation in the text to support them/ The authors are encouraged to carefully revise the text to make it more accessible and easier to understand to someone not directly within their field of study.

MINOR:

Abstract line 14, line 31 and elsewhere: Specify “DNA double-strand break”

Abstract and elsewhere - use past tense (e.g. line 17: combined, elevated, demonstrated, observed, measured, etc.)

Minor Line 73: Which phenotype? Inability of zip4 to be rescued by fancm? Please clarify text.

Minor: Line 75-79. This part is also confusing. The text appears to set the user up to expect a test of the fancm zip4 in an engineered Col/Ler hybrid, yet the double is only tested in the two inbred lines. Can this be explained more clearly why this wasn't done?

Minor: Line 80. “Is typically detected” sounds like it is an assay that is used rather than the biological mechanism that has evolved within cells. Can this be made clearer?

Minor: Line 89-103: Please explain the use of the Mann-Whitney vs Welch's t-test. Is there a rationale for these types of statistical tests? Why use different tests for the same measurement (CO frequency)? Yet more confusingly, fig 1g itself uses a Kruskal-Wallis H-test with a different P-value than is reported in the main text.

Minor : Line 110-111. Was the sequencing coverage similar for all samples? It seems that CO discovery will depend on depth.

Minor: Line 122-123. Why is this increase (upon MSH2 deletion) assumed to be class II COs? Why cannot it not be an increase in class I COs (based on the available evidence presented)?

Line 126: None of the lines show recombination “in” centromeres. Do the authors mean in the central region flanking the CEN? Fig S3 should also be referenced here—and in that figure, the plots labelled for which chromosome is which.

Line 127-129. How is the reader meant to appraise this statement when the two mutants being compared on different panels? This is especially problematic because the left-hand Y-axis changes scale across the relevant plots in Fig 2c.

Line 130. Even so...why does this suggest a similar activity of Class I and Class II COs? Please explain the logic that leads to this conclusion.

Line 131: Which panel(s) support an increase in COs at chromosome ends in the stated lines (relative to wild type)? Please refer to them. Also do the authors mean an increased proportion (of all detected COs), or an increase in absolute terms?

Major Line 132-134. Again: Please explain the logic that underpins this statement that MSH2 inactivation increases class II CO frequency? "indicate".

Line 145-146. Unhelpfully cryptic: What does "In the pathway controlled by RECQ4" mean? Does this mean the class I or class II pathway? (since the balance between the two pathways is affected by RECQ4 loss, the statement could mean either or both).

Please explain why you infer that the increase in CO number caused by MSH2 deletion in the recq4 mutant are class II (at least...that is what I think is meant.). Overall, this section would benefit from a much clearer explanation of how the data are interpreted.

Line 151-152. Again logic/info is missing here. For what reason do the authors conclude this? Is it due to the difference in polymorphism at the proximal versus telomeric ends of the chromosome? If so, please state this. If it is for some other reason, please state that.

Section 153-191. This section is interesting. I noticed that panel f-g may show a more complex trend with an apparent peak in the raw data around 3-4 SNPs/kb. The fitted model seems to not be sensitive enough to detect this potential trend. Perhaps fitting a smoothed average would be more meaningful. This may indicate that Msh2 has little effect when SNP density is very low (0-1 SNP/kb), nor when it is high >10 SNP/kb.

Line 196-199 require a reference (I assume the authors are not referring to data in this submission?). Also, are the authors referring to the effect (drastic decrease in het regions) just in zip4 or fancm single mutants, or the effect of MSH2 loss in zip4 or fancm? The authors really need to take care to be explicit in describing the observations that underpin their logic. The paper is observation dense and frequently extremely hard to follow.

Line 257-258. I struggled to follow this paragraph and then got especially lost in this line. What does "in the hybrid" refer to? Does it mean HOM-HET? If so, why is this not referred to in this same way to avoid reader confusion? Having said that...why is there no measure of recombination in the 3.9 interval in the HOM-HET orientation (in msh2) in fig 4f? The authors would do well to revise these section and seek external readers to critique how easy their experiments and analyses are to follow for non experts.

Line 283-284. To aid clarity, I suggest to change to: "...class II COs that form in the absence of FANCM cannot be..."

Line 293-294. It is unclear how Fig 5a helps explain what is the changed pattern of heterozygosity that is mentioned in the text. Changed from what to what?

Line 297. Which are the proximal regions? Do the authors mean CENtrome-re-proximal?

Reviewer #2 (Remarks to the Author):

The work from Dluzewska et al and the Ziolkowski team is valuable research on the relationship between polymorphism density and crossover pathways. This work should be published in my opinion after some considerations are addressed.

Major comments

-Regarding chiasma quantifications in figure 1f and crossover frequency in figure 1g. It is difficult to reconcile figure 1f and figure 1g. Figure 1g shows crossovers in interval 420 in *fanm zip4 msh2* to be well in excess of the wildtype frequency. Yet in figure 1f, the opposite is true. The authors acknowledge this on lines 105-106. I think that this suggests that chiasma quantification is not a reliable technique to be using, particularly for an analysis of class II crossover formation, which more often than not will result in two crossovers/chiasmata being relatively close to one another and beyond the detection limits of the cytological technique in figure 1f. No detail is given in figure 1f or supplementary table 4 about how many plants were used for this cytological analysis, which also raises issues about the broader reliability of data in figure 1f if it comes from one plant per group given the variables that can affect crossovers beyond genetics. Some of the authors have published previously on the effect of temperature on crossover frequency and positioning in plants, and the FTLs have been used to show that different branches/bolts and temperatures affect crossover frequencies (Francis et al, 2007, PNAS). The authors could consider being more conservative with the cytology and perhaps present the cytology as the number univalents and bivalents. Other than figure 1f, the rest of the paper is very strong with respect to the rigor of the replication and statistical methods used.

The authors show an informative analysis in Figure 3 of the relationship between snp density and crossover frequency. This could be subjectively improved in my opinion by including:
-a plot/figure of how big a block of homozygosity or heterozygosity needs to be in physical and/or genetic distance to trigger/suppress the effects described on crossover formation from the separate pathways. I.e. is it a 1Mb block of homozygosity, or more? Also, how does crossover detection capability play into this as a technical challenge, or do the authors have sufficient sequencing depth and power for this to be a non-issue?

-Figure 3 could perhaps be made more useful/easier to interpret to the average reader by putting which type of crossover they are analyzing. e.g. in *zip4 fanm msh2* it's "Class II Cos", in *Hei10-OE* it's "Class I COs".

_ I think that the titles of sections on lines 135 and 104, are unintentionally written in a way that is inaccurate. The authors probably meant to say "Class II crossovers are repressed by MSH2 in *RECQ4* mutants"? and "class II crossovers are repressed by MSH2 in *fanm* mutants"?

Minor comments

line 195 - "...depends on MSH2 heterodimers....". Was this shown directly with biochemistry, or inferred with genetic mutants?

lin385-387 - regarding concentration of MSH2 as a rate limiting factor. Is there any direct evidence that supports this with concentrations of MSH2 known from meiocytes? or even a representative arabidopsis somatic cell type like root tips?

Figure 2b - please increase the spacing between genotypes on the x-axis of the plots. It is not appropriate that data points overlap between genotypes.

Reviewer #3 (Remarks to the Author):

Dluzewska *et al.*, sought to elucidate the often-contradictory effects of DNA sequence polymorphisms on crossover formation in Arabidopsis. Through mutating MSH2, which controls DNA mismatch recognition, the authors found that DNA polymorphisms, both with and without MSH2 activity, inhibit class II CO formation. They also found that MSH2 stimulates class I CO formation and that there is a reorganization of COs depending on the number of polymorphisms and the location with respect to pericentromeric regions and high homozygosity regions. This manuscript is adding significant and novel knowledge to the field and has cleared up the confusing results from the past.

The main and supporting figures are clear, concise, easy to understand, and significantly improve the comprehension of the results. However, the manuscript text requires some serious changes to be made before it is published. Specifically, the results section is quite confusing and difficult to follow, due to the complexity of the experimental design. Furthermore, a few conclusions do not have valid or sufficient support from the data, and these extrapolations need to be stated more clearly.

Major changes:

1. Overall, there is a lot of data, and the manuscript is quite dense with information. It is often hard to follow the thought process. I would appreciate it if the results section was simplified extensively. Including a concise list of specific conclusions would help the readers to understand each finding more clearly.
2. It is possible that the extra class II COs generated in the *fancm* and *recq4* mutants are abnormal, i.e., not entirely the same as the class II COS found in wild type. If class II COs are rare to ensure genome stability, vastly increasing their numbers could lead to CO formation from recombination intermediates that would never lead to CO formation in wild type. The authors should be specific that their conclusion relates to *fancm/recq4* generated class II COs and that wild-type class II might behave differently.
3. The conclusions on the *fancm msh2* double mutant increasing only class II COs assumes that FANCM is only involved in the class II CO pathway. However, recent report suggests that FANCM may play a role in class I as class II CO formation (Li *et al.*, 2021 "Fanconi anemia ortholog FANCM regulates meiotic crossover distribution in plants" and Desjardins *et al.*, 2022 "FANCM promotes class I interfering crossovers and suppresses class II non-interfering crossovers in wheat meiosis"). The conclusion on lines 122-123 should be worded more carefully.
4. Line 72: Please add a sentence explaining what the *zip4* mutation does. Relevant information on the *zip4* mutation was not explained until the discussion.
5. Line 95-96: Please add a sentence explaining Col-420 reporter line. It is introduced without any additional information.

Minor changes:

- Line 30: It's 'crossover' not 'crossover recombination'.
Line 71: 'from us and others' is too informal.
Line 108: We investigated the crossover distribution.

Lines 124-126: Be more specific which aspect(s) of mutants/wild-type recombination profiles were strongly correlated.

Lines 195-199: Be specific that the effect of *zip4* was observed in inbred not hybrids.

Lines 209-210: What is 'very low' recombination frequency?

Line 267: In both mutants, not "both multiple mutants".

Line 302: 'Very' in front of "HET-HOM border" is not necessary.

Line 342: The statement that class II COs are 'unable to secure the proper course of meiosis' is confusing.

Dear Reviewers,

Our responses to your comments are highlighted in blue type.

Due to the journal policy, most of Supplementary Tables in the current version of the work has been moved to the Data Source file.

We have put a lot of effort into improving the manuscript, to make it more accessible to the readers. Since showing all the changes we made at the same time can make reading the manuscript much more difficult, we have included both the revised version and the version with track changes.

When we refer to specific lines in the text in our responses, we refer to the revised manuscript without marked changes.

Reviewer #1 (Remarks to the Author):

OVERVIEW

Recombination is critical to generate genetic diversity within sexually reproducing organisms, but how it is controlled and influenced by features such as DNA sequence polymorphisms that exist within sexually breeding species is unclear. The authors use *Arabidopsis* to explore the effect that the key mismatch repair protein, MSH2 has on recombination in meiosis—specifically in crossover formation—in various mutants that alter the relative proportion of class I and class II COs. The study is detailed and extensive, however, I found many statements and inferences to lack sufficient explanation in the text to support them/ The authors are encouraged to carefully revise the text to make it more accessible and easier to understand to someone not directly within their field of study.

Thank you for appreciating our work. The new version of the manuscript has been extensively revised to make it easier to understand, also by including comments from a non-expert within the field of meiotic recombination.

MINOR:

Abstract line 14, line 31 and elsewhere: Specify “DNA double-strand break”

Thanks for this comment!

Abstract and elsewhere - use past tense (e.g. line 17: combined, elevated, demonstrated, observed, measured, etc.)

We did not use the past tense because *Nature Communications* policy requires the use of the present tense in the abstract.

Minor Line 73: Which phenotype? Inability of *zip4* to be rescued by *fancm*? Please clarify text.

Clarified as requested.

Minor: Line 75-79. This part is also confusing. The text appears to set the user up to expect a test of the *fancm zip4* in an engineered Col/Ler hybrid, yet the double is only tested in the two inbred lines. Can this be explained more clearly why this wasn't done?

We agree that this paragraph may have been misleading, so we have corrected it. Since we did not have the *fancm zip4* double mutant for the *Ler* background, but only the *fancm* single mutant, we show in this paragraph that we generated such a mutant via CRISPR-Cas9. We then used it in a cross with Col *fancm zip4* to produce a Col/Ler *fancm zip4* hybrid.

Minor: Line 80. “Is typically detected” sounds like it is an assay that is used rather than the biological mechanism that has evolved within cells. Can this be made clearer?

We agree with this comment and corrected the text accordingly.

Minor: Line 89-103: Please explain the use of the Mann-Whitney vs Welch's t-test. Is there a rationale for these types of statistical tests? Why use different tests for the same measurement (CO frequency)? Yet more confusingly, fig 1g itself uses a Kruskal-Wallis H-test with a different P-value than is reported in the main text.

We thank the reviewer for this comment. For all crossover frequency measurements based on FTLs, when different lines or different mutants are compared, Welch's t-test was used, which is an adaptation of Student's t-test and is more reliable when the samples have unequal variances and possibly unequal sample sizes (Ruxton, 2006, doi:10.1093/beheco/ark016; Derrick et al., 2016, doi.org/10.20982/tqmp.12.1.p030). The only exception was the analysis along the chromosome arm, in which we compared changes in crossover frequency in eight adjacent FTL intervals (Fig. 5). To be able to correct for multiple comparisons and determine at which intervals changes are significant, we used One-way ANOVA with Tukey HSD (Tukey, 1949, doi.org/10.2307/3001913). Kruskal-Wallis H test followed by Mann Whitney U test with Bonferroni correction was used to compare the number of crossovers per F₂ individual between different populations (Fig. 2b), because in this case we had a large number of samples per genotype (>170). In the case of Fig. 1g, Welch's t-test was used, but the caption for Fig. 2b was mistakenly copied from Fig. 2b. We apologize for this mistake, which is now corrected.

Minor : Line 110-111. Was the sequencing coverage similar for all samples? It seems that CO discovery will depend on depth.

Sequencing coverage of individual samples in a particular pool varies due to different DNA concentration. However, according to Rowan et al. 2015, doi.org/10.1534/g3.114.016501, where the TIGER algorithm for crossover detection was described, even at 0.1x coverage, 97.5% of crossovers are detected, increasing to 99.3% at 10x coverage. To further improve CO detection, we applied filtering to keep samples representing at least 100,000 reads: with a read length of 2x150 bp and a reference genome size of 115 Mbp, this gives a minimum sequencing coverage of 0.26x. Therefore, we are convinced that we identify the vast majority of crossovers in our analysis.

Some details on the sequencing libraries prepared in this work are provided in Supplementary Table 3.

Minor: Line 122-123. Why is this increase (upon MSH2 deletion) assumed to be class II COs? Why cannot it not be an increase in class I COs (based on the available evidence presented)?

Various analyzes in the original work by Crismani et al. 2012 doi.org/10.1126/science.1220381, including a comparison of the number of MLH1 foci between the wild type and *fancm*, indicate that in the *fancm* mutant of *A. thaliana* the Class I crossover number is not affected. Given that the number of Class I crossovers decreases minimally in the *msh2* single mutant, we do not consider it likely that there would be an increase in Class I crossovers in the *fancm zip4* double mutant. However, crossovers in the ZMM pathway may be redistributed in this mutant (Li et al., 2021, doi.org/10.1093/plphys/kiab061; Desjardins et al., 2022, doi.org/10.1038/s41467-022-31438-6). Therefore, we rephrased this conclusion to read: "Therefore, it can be assumed that the inactivation of MSH2 in *fancm* also leads to an increase in Class II crossovers (Fig. 2b), although effects of FANCM inactivation on Class I crossover distributions cannot be excluded^{25,65}."

Line 126: None of the lines show recombination "in" centromeres. Do the authors mean in the central region flanking the CEN? Fig S3 should also be referenced here—and in that figure, the plots labelled for which chromosome is which.

Thanks for pointing this out! We have replaced "centromeres" with the more correct expression "pericentromeric regions". We also added chromosome numbers to Supplementary Fig. 4.

Line 127-129. How is the reader meant to appraise this statement when the two mutants being compared on different panels? This is especially problematic because the left-hand Y-axis changes scale across the relevant plots in Fig 2c.

We were referring here to Fig. 2c, which shows the correlation coefficient matrices between the crossover landscape observed in 0.3 Mb adjacent windows for different genotypes. However, to help the reader compare the profile and recombination frequency in the *msh2 fancm* mutant against the wild type and *msh2 fancm zip4* mutant, we have added additional panels in both Fig. 2 and Supplementary Fig. 4 to show these three genotypes on one scale (i.e., wild type, *msh2 fancm* and *msh2 fancm zip4*).

Line 130. Even so...why does this suggest a similar activity of Class I and Class II COs? Please explain the logic that leads to this conclusion.

We agree with the reviewer that in this case we are probably drawing too far-reaching conclusions about the relative activity of the two crossover pathways. Therefore, we have decided to remove this sentence.

Line 131: Which panel(s) support an increase in COs at chromosome ends in the stated lines (relative to wild type)? Please refer to them. Also do the authors mean an increased proportion (of all detected COs), or an increase in absolute terms?

We have added references to the newly added panels in Fig. 2c,d and Supplementary Fig. 4 where all three genotypes i.e. wild type, *msh2 fancm* and *msh2 fancm zip4* are shown together. We also specified in the text that we are referring here rather to an increased proportion of COs in the subtelomeric regions, because the absolute values of COs for the *msh2 fancm zip4* genotype may be overestimated due to reduced fertility. Information on the potential overestimation of the number of COs in *zip4* mutants is included in line 129-132 and also in the Fig. 2 caption (line 882-884).

Major Line 132-134. Again: Please explain the logic that underpins this statement that MSH2 inactivation increases class II CO frequency? "indicate".

Since only Class II crossovers can be formed in the *zip4* mutant, the increase in the crossover frequency in *msh2 fancm zip4* relative to *fancm zip4* can only be caused by an increase in the number of class II events. For clarity, we have added a sentence explaining this point at the very beginning of Results section (line 84).

Line 145-146. Unhelpfully cryptic: What does "In the pathway controlled by RECQ4" mean? Does this mean the class I or class II pathway? (since the balance between the two pathways is affected by RECQ4 loss, the statement could mean either or both).

Please explain why you infer that the increase in CO number caused by MSH2 deletion in the *recq4* mutant are class II (at least...that is what I think is meant.). Overall, this section would benefit from a much clearer explanation of how the data are interpreted.

We agree that this statement was imprecise, so it has been rephrased to read: "This suggests that MSH2 represses Class II crossover formation in response to interhomolog polymorphism in the pathway inhibited by RECQ4 helicase." In the first part of this paragraph, we also explained that there is an increase in non-interfering Class II crossovers in the *recq4* mutant (which is extensively documented by others, as referenced in the manuscript). We hope that now the whole section is clearer and easier to follow.

Line 151-152. Again logic/info is missing here. For what reason do the authors conclude this? Is it due to the difference in polymorphism at the proximal versus telomeric ends of the chromosome? If so, please state this. If it is for some other reason, please state that.

Unfortunately, our data do not allow us to unequivocally conclude whether the smaller increase in recombination in *msh2 recq4* compared to *recq4* in pericentromeres is due to a higher density of polymorphisms or maybe some pericentromeric region-associated epigenetic factors. Therefore, we decided to limit ourselves to the statement that "MSH2-dependent polymorphism detection has a limited effect on Class II crossover formation in pericentromeres".

Section 153-191. This section is interesting. I noticed that panel f-g may show a more complex trend with an apparent peak in the raw data around 3-4 SNPs/kb. The fitted model seems to not be sensitive enough to detect this potential trend. Perhaps fitting a smoothed average would be more meaningful. This may indicate that Msh2 has little effect when SNP density is very low (0-1 SNP/kb), nor when it is high >10 SNP/kb.

We thank the reviewer for this helpful comment. We reanalyzed our data using the Local Polynomial Regression Fitting (loess) to plot trend lines, which allowed us to capture more subtle effects in the relationships between SNP density and crossover frequency (Fig. 3). The part of the manuscript referring to these analyses has been rewritten, also in terms of clarity of expression. We are confident that this version will meet the reviewer's expectations.

Line 196-199 require a reference (I assume the authors are not referring to data in this submission?). Also, are the authors referring to the effect (drastic decrease in het regions) just in *zip4* or *fancm* single mutants, or the effect of MSH2 loss in *zip4* or *fancm*? The authors really need to take care to be explicit in describing the observations that underpin their logic. The paper is observation dense and frequently extremely hard to follow.

We apologize for the lack of precision. We rephrased this fragment and added a reference.

Line 257-258. I struggled to follow this paragraph and then got especially lost in this line. What does "in the hybrid" refer to? Does it mean HOM-HET? If so, why is this not referred to in this same way to avoid reader confusion? Having said that...why is there no measure of recombination in the 3.9 interval in the HOM-HET orientation (in *msh2*) in fig 4f? The authors would do well to revise these section and seek external readers to critique how easy their experiments and analyses are to follow for non experts.

As suggested by the reviewer, we unified the text and verified its clarity. We have added an index to names of the lines with a different heterozygosity pattern, specifying the location of the interval used to measure CO frequency, so that it is easier to find out whether the measured interval is in the heterozygous or homozygous state in a

given line (e.g. HET⁴²⁰-HOM, HET-HOM^{3.9}). We hope that the new version of the manuscript will be easier to follow.

In the case of the 3.9 interval, we have no technical possibility to measure the recombination in the HOM-HET system, because the HOM-HET line already carries fluorescent reporters expressed in seeds in the 420 region (we have no possibility to measure the recombination at the 420 and 3.9 intervals at the same time). For these reasons, we supplemented our results with the *CEN3* interval, which is based on segregation of fluorescent reporters expressed in pollen, but in this case we did not have *fancm zip4* and *msh2 fancm zip4* mutants.

Line 283-284. To aid clarity, I suggest to change to: "...class II COs that form in the absence of FANCM cannot be...

Thanks for this useful suggestion, which we also adopted to specify Class II crossovers in the titles of earlier sections.

Line 293-294. It is unclear how Fig 5a helps explain what is the changed pattern of heterozygosity that is mentioned in the text. Changed from what to what?

This sentence has been rephrased (line 355-357).

Line 297. Which are the proximal regions? Do the authors mean CENtrome-proximal?

We corrected as recommended.

Reviewer #2 (Remarks to the Author):

The work from Dluzewska et al and the Ziolkowski team is valuable research on the relationship between polymorphism density and crossover pathways. This work should be published in my opinion after some considerations are addressed.

Thank you for the kind words!

Major comments

-Regarding chiasma quantifications in figure 1f and crossover frequency in figure 1g. It is difficult to reconcile figure 1f and figure 1g. Figure 1g shows crossovers in interval 420 in *fancm zip4 msh2* to be well in excess of the wildtype frequency. Yet in figure 1f, the opposite is true. The authors acknowledge this on lines 105-106. I think that this suggests that chiasma quantification is not a reliable technique to be using, particularly for an analysis of class II crossover formation, which more often than not will result in two crossovers/chiasmata being relatively close to one another and beyond the detection limits of the cytological technique in figure 1f.

The reviewer is correct that there is a difference between crossover counts in the two different techniques. However, both techniques show that mutating *MSH2* increases crossovers in a *fancm zip4* mutant Col/*Ler* background, suggesting that *MSH2* is suppressing Class II crossovers in a non-homozygous (hybrid) background. Therefore the pattern is the same, but the numbers are different, so why would this be? There are two reasons for this: First, as we explained in the previous paragraph (line 129-132), the scarcity of crossovers in *zip4* and *fancm zip4* results in the formation of univalents (Fig. 1e,f) and consequently the formation of unbalanced and usually non-functional gametes (Mercier et al., 2015, doi.org/10.1146/annurev-arplant-050213-035923). Seeds therefore arise primarily from those gametes that have experienced enough crossover events for proper chromosome segregation. Consequently, crossover frequency measurements based on fluorescent seeds show inflated recombination values. Such a problem does not occur in cytological analysis, where chiasmata are counted in all meiocytes, including those which due to incorrect chromosome segregation caused by the lack of crossover will not produce functional gametes. Second, in the seed scoring analysis, we measure the frequency of crossovers only at a specific interval on the chromosome - in this case, it is the subtelomeric region of chromosome 3 (interval 420), which has a relatively low polymorphism density between Col and *Ler*. We cannot conclude from this analysis what happens in the interstitial or pericentromeric regions, which have much higher levels of Col/*Ler* polymorphisms (see gray shaded plot on Fig. 2e or Suppl. Fig. 4). Cytological analysis, in turn, shows ALL chiasmata, regardless of their chromosomal location. Suspecting that in more polymorphic regions the number of crossovers in the tested mutants is much lower, we decided to perform GBS analysis, where we can see all crossovers with high resolution, regardless of the chromosomal location. We have tried to highlight this issue in the revised manuscript. This analysis confirmed our supposition regarding the uneven crossover distribution in *fancm zip4* and *msh2 fancm zip4* (Fig. 2e). Coming back to the potential underestimation of chiasmata counts in cytological analysis, this does not seem likely to us. Two crossovers close together may be scored as one chiasma, but with such a low number of

chiasmata per cell the counts will be accurate, although there is an expectation they could be slightly underscored (but not to account for the 420 levels). The cytological data is consistent with a model in which *fancm* does not restore crossovers to wild type levels in Col/Ler, whereas it does in a Col/Col background (Knoll et al. 2012; Crismani et al. 2012). By knocking out *MSH2*, more chiasmata are able to form, but as the reviewer points out, not to the scale of the 420 interval. However, the observed phenomenon is the same regardless of the used analysis.

No detail is given in figure 1f or supplementary table 4 about how many plants were used for this cytological analysis, which also raises issues about the broader reliability of data in figure 1f if it comes from one plant per group given the variables that can affect crossovers beyond genetics. Some of the authors have published previously on the effect of temperature on crossover frequency and positioning in plants, and the FTLs have been used to show that different branches/bolts and temperatures affect crossover frequencies (Francis et al, 2007, PNAS).

We apologize for omitting this information. Genotypes were grown together and primary inflorescences were fixed in 3:1 ethanol:acetic at the same time from 3 individual plants and 20 cells per plant were imaged and analysed. We have supplemented this information in both Methods and the figure caption. Detailed data on the chiasma counts in individual cells are included in the Source Data file.

The authors could consider being more conservative with the cytology and perhaps present the cytology as the number univalents and bivalents. Other than figure 1f, the rest of the paper is very strong with respect to the rigor of the replication and statistical methods used.

As suggested by the reviewer, in the revised manuscript we presented the results as rods, rings and univalents (Fig. 1f). We moved the estimated chiasma number plot to the Supplementary Information (new Supplementary Fig. 3).

The authors show an informative analysis in Figure 3 of the relationship between snp density and crossover frequency. This could be subjectively improved in my opinion by including:
-a plot/figure of how big a block of homozygosity or heterozygosity needs to be in physical and/or genetic distance to trigger/suppress the effects described on crossover formation from the separate pathways. I.e. is it a 1Mb block of homozygosity, or more?

The reviewer raises here an interesting issue of the importance of the size of heterozygous or homozygous blocks, at which the effect of stimulation or suppression of recombination in individual crossover pathways is triggered. Unfortunately, the data shown in Fig. 3 reflects meiosis in F₁ individuals in which the entire genome is heterozygous. Although the distribution of polymorphism along the chromosome is not uniform and it is possible to distinguish highly polymorphic regions and polymorphism-free regions, the transition from one to the other is usually smooth and depends largely on the location on the chromosome. This makes it impossible to perform the analysis requested by the reviewer.

Although in this work we did not analyze the minimum length of a heterozygous block that causes local crossover stimulation in a homozygous background, in our recently published paper we showed that the effect is statistically significant already for a block of only 3.3 kb containing 44 SNPs (Szymanska-Lejman et al., 2023, doi.org/10.1038/s41467-022-35722-3). This is at the scale of a single crossover hotspot showing that the effect operates at high resolution. These data were obtained for wild-type lines and their counterparts in the *msh2* background, so they mainly concern Class I crossovers. At the moment, we do not have data that could show what is the minimum length of a heterozygous (or homozygous) block causing the effect of changing the Class II crossover frequency.

Also, how does crossover detection capability play into this as a technical challenge, or do the authors have sufficient sequencing depth and power for this to be a non-issue?

In all data presented in the work, crossover mapping was performed using the TIGER pipeline, which is a high-performance method for identifying crossover sites (Rowan et al., 2015, doi.org/10.1534/g3.114.016501). According to the authors, even at 0.1× coverage, 97.5% of crossovers are detected, increasing to 99.3% at 10× coverage. In addition, for all the samples we analyzed, we applied the rule that the total number of mapped reads for a sample cannot be less than 100,000 reads, which, with a read length of 2×150 bp and a reference genome size of 115 Mbp, gives a minimum sequencing coverage of 0.26×. Therefore, we are confident that our crossover detection is very high and should not be an issue.

The average crossover mapping resolution, defined as the distance between two SNP markers between which there is a crossover, ranged for different genotypes from 0.7 kb to 1.7 kb (median size between 358 to 439 bp). Given that the analysis in Fig. 3 was performed in 100 kb non-overlapping windows, this resolution does not appear to affect the results obtained.

-Figure 3 could perhaps be made more useful/easier to interpret to the average reader by putting which type of crossover they are analyzing. e.g. in zip4 fancm msh2 it's "Class II Cos", in Hei10-OE it's "Class I COs".

Thank you for this suggestion which we find very helpful. We have added information on the percentage share of Class I and II crossover in the analyzed mutants in Fig. 3.

_ I think that the titles of sections on lines 135 and 104, are unintentionally written in a way that is inaccurate. The authors probably meant to say "Class II crossovers are repressed by MSH2 in RECQ4 mutants"? and "class II crossovers are repressed by MSH2 in fancm mutants"?

Thank you for this comment. Indeed, the section names were inaccurate, so we followed those suggestions.

Minor comments

line 195 - "...depends on MSH2 heterodimers....". Was this shown directly with biochemistry, or inferred with genetic mutants?

This was inferred from the analysis of genetic mutants. We have added this information in the revised manuscript (line 247-249).

lin385-387 - regarding concentration of MSH2 as a rate limiting factor. Is there any direct evidence that supports this with concentrations of MSH2 known from meiocytes? or even a representative arabidopsis somatic cell type like root tips?

We admit that the presented hypothesis is very speculative and we are not aware of any data that would allow to demonstrate the limited availability of MSH2 complexes in meiosis. The level of *MSH2* transcript in meiocytes is 7.14 times higher than in leaves (Walker et al., 2017; doi.org/10.1038/s41588-017-0008-5), which seems a relatively small increase when compared to other repair proteins that function in somatic cells. Moreover, it is expressed at relatively low level. The table below, constructed based on the data from the abovementioned publication, shows the transcript levels of various repair proteins that function in both meiosis and somatic cells, ranked from the lowest meiocyte/leaf ratio; MSH2 is marked in orange. Interestingly, the other proteins included in the MSH2 heterodimers (highlighted in light green) show much higher levels of expression, which may indicate that MSH2 is the limiting factor.

Gene	Name	Leaf_RNAseq	Meiocyte_RNAseq_	Meiocytes/leaf	Function
AT1G78790	MHF2	0.62	4.39	7.1	FANCM complex
AT3G18524	MSH2	0.25	1.79	7.1	MSH2 dimers
AT4G09140	MLH1	0.30	5.41	17.9	MutL dimers
AT1G10930	RECQ4A	0.36	8.59	23.7	RECQ4 complex
AT3G02680	NBS	0.11	2.89	25.9	MRN complex
AT4G02460	PMS1	0.18	4.73	26.6	MutL dimers
AT1G01880	GEN1	0.10	2.55	26.8	Structure-specific nuclease
AT4G02070	MSH6	0.19	5.14	27.2	MSH2 dimers
AT3G24495	MSH7	0.15	5.53	36.9	MSH2 dimers
AT2G31970	RAD50	0.83	36.84	44.4	MRN complex
AT5G54260	MRE11	0.15	8.40	54.6	MRN complex
AT1G35530	FANCM	0.06	3.68	62.9	FANCM complex
AT1G60930	RECQ4B	0.11	9.08	85.5	RECQ4 complex
AT4G30870	MUS81	0.05	7.21	144.7	MUS81 complex
AT2G21800	EME1A	0.01	2.46	221.1	MUS81 complex
AT5G50930	MHF1	0.04	10.20	242.3	FANCM complex
AT5G20850	RAD51	0.03	9.47	301.2	Recombinase
AT5G63540	RMI1	0.00	1.70	356.9	RECQ4 complex
AT2G22140	EME1B	0.03	13.21	394.0	MUS81 complex
AT5G63920	TOP3A	0.04	18.84	434.0	RECQ4 complex

AT3G48900	SEND1	0.01	3.90	644.4	Structure-specific nuclease
AT4G25540	MSH3	0.01	7.57	1460.4	MSH2 dimers

However, we do not have data at the protein level, nor do we know how much of the protein remains in the cytoplasm.

In the perfect world, we would compare the distribution of MSH2 along the chromosomes in the HET-HOM and HOM-HET lines in prophase I. Unfortunately, the quality of antibodies available for MSH2 is very poor and not suitable for such analyzes (Blackwell et al., 2020, doi.org/10.15252/embj.2020104858). Therefore, we want to keep our speculative model in the discussion, hoping that in the future it will be possible to revise it with new results.

Figure 2b - please increase the spacing between genotypes on the x-axis of the plots. It is not appropriate that data points overlap between genotypes.

Corrected as requested.

Reviewer #3 (Remarks to the Author):

Dluzewska *et al.*, sought to elucidate the often-contradictory effects of DNA sequence polymorphisms on crossover formation in Arabidopsis. Through mutating MSH2, which controls DNA mismatch recognition, the authors found that DNA polymorphisms, both with and without MSH2 activity, inhibit class II CO formation. They also found that MSH2 stimulates class I CO formation and that there is a reorganization of COs depending on the number of polymorphisms and the location with respect to pericentromeric regions and high homozygosity regions. This manuscript is adding significant and novel knowledge to the field and has cleared up the confusing results from the past.

We thank the reviewer for appreciating our work!

The main and supporting figures are clear, concise, easy to understand, and significantly improve the comprehension of the results. However, the manuscript text requires some serious changes to be made before it is published. Specifically, the results section is quite confusing and difficult to follow, due to the complexity of the experimental design. Furthermore, a few conclusions do not have valid or sufficient support from the data, and these extrapolations need to be stated more clearly.

Major changes:

1. Overall, there is a lot of data, and the manuscript is quite dense with information. It is often hard to follow the thought process. I would appreciate it if the results section was simplified extensively. Including a concise list of specific conclusions would help the readers to understand each finding more clearly.

The manuscript has been significantly improved for clarity, and we have added conclusions at the end of each paragraph (where possible) to make it easier for the reader to follow.

2. It is possible that the extra class II COs generated in the *fancm* and *recq4* mutants are abnormal, i.e., not entirely the same as the class II COS found in wild type. If class II COs are rare to ensure genome stability, vastly increasing their numbers could lead to CO formation from recombination intermediates that would never lead to CO formation in wild type. The authors should be specific that their conclusion relates to *fancm/recq4* generated class II COs and that wild-type class II might behave differently.

This is an important comment from the reviewer, with which we fully agree. We have carefully reviewed the entire manuscript and, where necessary, have clarified that the observed Class II crossovers are formed in the unnatural conditions of the absence of DNA helicases. We have also underlined this fact at the beginning of the discussion (line 411-413).

3. The conclusions on the *fancm msh2* double mutant increasing only class II COs assumes that FANCM is only involved in the class II CO pathway. However, recent report suggests that FANCM may play a role in class I as class II CO formation (Li *et al.*, 2021 "Fanconi anemia ortholog FANCM regulates meiotic crossover distribution in plants" and Desjardins *et al.*, 2022 "FANCM promotes class I interfering crossovers and suppresses class II non-interfering crossovers in wheat meiosis"). The conclusion on lines 122-123 should be worded more carefully.

Various analyzes in the original work by Crismani et al., 2012, doi.org/10.1126/science.1220381, including a comparison of the number of MLH1 foci between wild type and *fancm* in *A. thaliana*, indicate that in *fancm* the

Class I crossover number is not affected. Given that the number of Class I crossovers decreases minimally in the *msh2* single mutant, we do not consider it likely that there would be an increase in Class I crossovers in the *fancm zip4* double mutant. However, we agree with the reviewer that crossovers in the ZMM pathway may be redistributed in this mutant. Therefore, we rephrased this conclusion and also cited the above-mentioned publications to read: "Therefore, it can be assumed that the inactivation of MSH2 in *fancm* also leads to an increase in Class II crossovers (Fig. 2b), although effects of FANCM inactivation on Class I crossover distributions cannot be excluded^{25,65}."

4. Line 72: Please add a sentence explaining what the *zip4* mutation does. Relevant information on the *zip4* mutation was not explained until the discussion.

We added the relevant information as requested at the very beginning of the Results (line 84-86).

5. Line 95-96: Please add a sentence explaining Col-420 reporter line. It is introduced without any additional information.

We have added an appropriate explanation regarding the Col-420 line (line 125-127). Moreover, we have included an additional figure explaining the measurement of crossover frequency in the fluorescent seed system to help the reader to understand this system (Fig. 1h).

Minor changes:

Line 30: It's 'crossover' not 'crossover recombination'.

Corrected as requested.

Line 71: 'from us and others' is too informal.

We removed this part leaving "Earlier studies have shown...".

Line 108: We investigated the crossover distribution.

Corrected.

Lines 124-126: Be more specific which aspect(s) of mutants/wild-type recombination profiles were strongly correlated.

Rephrased.

Lines 195-199: Be specific that the effect of *zip4* was observed in inbred not hybrids.

Rephrased.

Lines 209-210: What is 'very low' recombination frequency?

We added specific values to the text.

Line 267: In both mutants, not "both multiple mutants".

Corrected.

Line 302: 'Very' in front of "HET-HOM border" is not necessary.

Corrected.

Line 342: The statement that class II COs are 'unable to secure the proper course of meiosis' is confusing.

We changed the sentence to read: "In wild type *A. thaliana*, the ZMM pathway that creates Class I crossover is dominant, while Class II crossovers are rare and therefore insufficient to secure balanced segregation of chromosomes in meiosis^{74,75}."

REVIEWERS' COMMENTS

Reviewer #1 (Remarks to the Author):

The authors have made a great number of changes to the description of their work, which, whilst still very information dense, is now easier to follow. They have also adequately answered the queries I raised in their specific rebuttal comments, which I assume will be available to readers. This is an extensive and interesting study very worthy of publication.

Reviewer #2 (Remarks to the Author):

The authors have satisfactorily addressed my concerns.

Reviewer #3 (Remarks to the Author):

The revised version of the manuscript has much more clear and concise writing, especially in the Results section. I believe the extrapolations used, that I raised concern about, have been better explained such as those on lines 160-162 and lines 411-413.

This manuscript, as mentioned in the first review, gives great insight into how polymorphisms influence the mechanisms of meiotic crossing-over. Because of the novelty of findings and of methods, with the new revised manuscript, it should be published.

Dear reviewers,

Our responses to your comments are highlighted in blue type.

Reviewer #1 (Remarks to the Author):

The authors have made a great number of changes to the description of their work, which, whilst still very information dense, is now easier to follow. They have also adequately answered the queries I raised in their specific rebuttal comments, which I assume will be available to readers. This is an extensive and interesting study very worthy of publication.

Reviewer #2 (Remarks to the Author):

The authors have satisfactorily addressed my concerns.

Reviewer #3 (Remarks to the Author):

The revised version of the manuscript has much more clear and concise writing, especially in the Results section. I believe the extrapolations used, that I raised concern about, have been better explained such as those on lines 160-162 and lines 411-413.

This manuscript, as mentioned in the first review, gives great insight into how polymorphisms influence the mechanisms of meiotic crossing-over. Because of the novelty of findings and of methods, with the new revised manuscript, it should be published.

We thank the reviewers for taking the time to re-examine our paper and we are pleased that all reviewers are satisfied with the changes we have made to our revised manuscript.